# DEJA VU: CONTINUAL MODEL GENERALIZATION FOR UNSEEN DOMAINS

**Chenxi Liu**[1*], **Lixu Wang**[1*‡†], **Lingjuan Lyu**[2‡], **Chen Sun**[2], **Xiao Wang**[1], **Qi Zhu**[1]
[1]Northwestern University, [2]Sony AI
{chenxiliu2020,lixuwang2025}@u.northwestern.edu,
{Lingjuan.Lv,chen.sun}@sony.com, {wangxiao,qzhu}@northwestern.edu

## ABSTRACT

In real-world applications, deep learning models often run in non-stationary environments where the target data distribution continually shifts over time. There have been numerous domain adaptation (DA) methods in both online and offline modes to improve cross-domain adaptation ability. However, these DA methods typically only provide good performance *after* a long period of adaptation, and perform poorly on new domains *before and during* adaptation – in what we call the *"Unfamiliar Period"*, especially when domain shifts happen suddenly and significantly. On the other hand, domain generalization (DG) methods have been proposed to improve the model generalization ability on unadapted domains. However, existing DG works are ineffective for continually changing domains due to severe catastrophic forgetting of learned knowledge. To overcome these limitations of DA and DG in handling the *Unfamiliar Period* during continual domain shift, we propose `RaTP`, a framework that focuses on improving models' target domain generalization (TDG) capability, while also achieving effective target domain adaptation (TDA) capability right after training on certain domains and forgetting alleviation (FA) capability on past domains. `RaTP` includes a training-free data augmentation module to prepare data for TDG, a novel pseudo-labeling mechanism to provide reliable supervision for TDA, and a prototype contrastive alignment algorithm to align different domains for achieving TDG, TDA and FA. Extensive experiments on Digits, PACS, and DomainNet demonstrate that `RaTP` significantly outperforms state-of-the-art works from Continual DA, Source-Free DA, Test-Time/Online DA, Single DG, Multiple DG and Unified DA&DG in TDG, and achieves comparable TDA and FA capabilities.

## 1 INTRODUCTION

A major concern in applying deep learning models to real-world applications is whether they are able to deal with environmental changes over time, which present significant challenges with data distribution shifts. When the shift is small, deep learning models may be able to handle it because their robustness is often evaluated and improved before deployment. However, when the data distribution shifts significantly for a period of time, in what we call the *"Unfamiliar Period"*, model performance on new scenarios could deteriorate to a much lower level. For example, surveillance cameras used for environmental monitoring can work normally with excellent performance on clear days, but have inferior performance or even become "blind" when the weather turns bad or the lighting conditions become poor (Bak et al., 2018). As another example, consider conducting lung imaging analysis for corona-viruses, deep learning models may present excellent performance after being trained on a large number of samples for certain variant (e.g., the Alpha variant of COVID-19), but are difficult to provide accurate and timely analysis for later variants (e.g., the Delta or Omicron variant) and future types of corona-viruses (Singh et al., 2020) when they *just appear*. In the following, we will first discuss related works, highlight their limitations in addressing the poor model performance during the *Unfamiliar Period*, and then introduce our approach.

---

[*]Equal contributions (ordered alphabetically); [‡]Corresponding authors.
[†]Part of the work was done during an internship at Sony AI.

**Domain adaptation (DA)** methods have been proposed to tackle continual data drifts in dynamic environments in either online or offline mode. For example, Continual DA (Liu et al., 2020; Rostami, 2021) starts from a labeled source domain and continually adapts the model to various target domains, while keeping the model performance from degrading significantly on seen domains. However, existing Continual DA works often assume that the source domain can be accessed all the time, which may be difficult to guarantee in practical scenarios, especially considering the possible limitation on memory storage and regulations on privacy or intellectual property. Source-Free DA (Yang et al., 2021; Qu et al., 2022) can overcome this issue and achieve target adaptation without the source domain data. In addition, Test-Time or Online DA (Wang et al., 2022; Iwasawa & Matsuo, 2021; Panagiotakopoulos et al., 2022) can improve the target model performance with a small training cost; however, the target domain data is only learned once by the model and the performance improvement is limited (higher improvement would require a large amount of data). With these DA methods, although the model may perform better on the new target domain *after* sufficient adaptation, its performance on the target domain *before and during* the adaptation process, i.e., in the *Unfamiliar Period*, is often poor. In cases where the domain shift is sudden and the duration of seeing a new target domain is short, this problem becomes even more severe. In this work, we believe that for many applications, it is very important to ensure that the model can also perform reasonably well in the *Unfamiliar Period*, i.e., before seeing a lot of target domain data. For instance in environmental surveillance, having poor performance under uncommon/unfamiliar weather or lighting conditions may cause significant security and safety risks. In the example of lung imaging analysis for corona-viruses, being able to quickly provide good performance for detecting new variants is critical for the early containment and treatment of the disease.

**Domain generalization (DG)** methods also solve the learning problem on multiple data domains, especially for cases where the target domain is unavailable or unknown during training. However, existing DG works are typically based on accurate supervision knowledge of the source domain data, whether it is drawn from a single domain (Wang et al., 2021c; Li et al., 2021) or multiple domains (Yao et al., 2022; Zhang et al., 2022), which may not be achievable in continually changing scenarios. Moreover, when DG is applied in scenarios with continual domain shifts, as it focuses more on the target domain, there could be severe catastrophic forgetting of domains that have been learned. There are also some works unifying DA and DG (Ghifary et al., 2016; Motiian et al., 2017; Jin et al., 2021); however they can only be used in standard DA or DG individually, thus still suffering their limitations. Bai et al. (2022) and Nasery et al. (2021) study the smooth temporal shifts of data distribution, but they cannot handle large domain shifts over time.

**Our Approach and Contribution.** In this work, we focus on the study of Continual Domain Shift Learning (CDSL) problem, in which the learning model is first trained on a labeled source domain and then faces a series of unlabeled target domains that appear continually. Our goal, in particular, is to improve model performance *before and during* the training stage of each previously-unseen target domain (i.e., in the *Unfamiliar Period*), while also maintaining good performance in the time periods *after* the training. Thus, we propose a framework called `RaTP` that optimizes three objectives: (1) to improve the model generalization performance on a new target domain *before and during* its training, namely the *target domain generalization* (TDG) performance, (2) to provide good model performance on a target domain *right after* its training, namely the *target domain adaptation* (TDA) performance, and (3) to maintain good performance on a trained domain *after* the model is trained with other domains, namely the *forgetting alleviation* (FA) performance. For improving TDG, `RaTP` includes a training-free data augmentation module that is based on Random Mixup, and this module can generate data outside of the current training domain. For TDA, `RaTP` includes a Top$^2$ Pseudo Labeling mechanism that lays more emphasis on samples with a higher possibility of correct classification, which can produce more accurate pseudo labels. Finally, for optimizing the model towards TDG, TDA, and FA at the same time, `RaTP` includes a Prototype Contrastive Alignment algorithm. Comprehensive experiments and ablation studies on Digits, PACS, and DomainNet demonstrate that `RaTP` can significantly outperform state-of-the-art works in TDG, including Continual DA, Source-Free DA, Test-Time/Online DA, Single DG, Multiple DG, and Unified DA&DG. `RaTP` can also produce comparable performance in TDA and FA as these baselines. In summary:

- We tackle an important problem in practical scenarios with continual domain shifts, i.e., to improve model performance *before and during* training on a new target domain, in what we call the *Unfamiliar Period*. And we also try to achieve good model performance *after* training, providing the model with capabilities of target domain adaptation and forgetting alleviation.

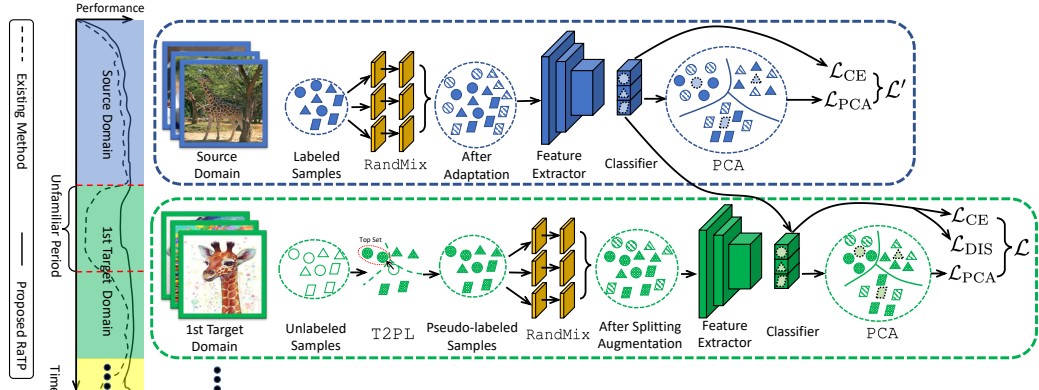

**Figure 1: Overview of applying our framework `RaTP` for Continual Domain Shift Learning (CDSL).**
`RaTP` first starts with a labeled source domain, applies `RandMix` on the full set of source data to generate augmentation data, and uses a simplified version $\mathcal{L}'$ of `PCA` for model optimization. Then, for continually arriving target domains, `RaTP` uses `T2PL` to generate pseudo labels for all unlabeled samples, applies `RandMix` on a top subset of these samples based on their softmax confidence, and optimizes the model by `PCA`.

- We propose a novel framework `RaTP` to achieve our goal. The framework includes a training-free data augmentation module that generates more data for improving model's generalization ability, a new pseudo-labeling mechanism to provide more accurate labels for domain adaptation, and a prototype contrastive alignment algorithm that effectively aligns domains to simultaneously improve the generalization ability and achieve target domain adaptation and forgetting alleviation.

- We conducted extensive experiments and comprehensive ablation studies that demonstrate the advantages of our `RaTP` framework over a number of state-of-the-art baseline methods.

## 2 METHODOLOGY

We first formulate the problem of Continual Domain Shift Learning (CDSL) in Section 2.1, and then introduce the three major modules of our framework `RaTP`. Specifically, Section 2.2 presents a Random Mixup data augmentation module `RandMix` that generates data for improving the target domain generalization (TDG) on unadapted domains, which is a crucial technique for improving model generalization in the *Unfamiliar Period*. Section 2.3 presents a Top$^2$ Pseudo Labeling approach `T2PL` that provides accurate labels to achieve target domain adaptation (TDA) for continually arriving target domains. Finally, Section 2.4 presents a Prototype Contrastive Alignment algorithm `PCA` that optimizes the model to achieve TDG, TDA, and forgetting alleviation (FA) on seen domains. Figure 1 shows the overall pipeline of `RaTP`.

### 2.1 PROBLEM FORMULATION OF CONTINUAL DOMAIN SHIFT LEARNING

We assume that there is a labeled source domain dataset $\mathcal{S} = \{(\boldsymbol{x}_i, \boldsymbol{y}_i) \| \boldsymbol{x}_i \sim \mathcal{P}_X^{\mathcal{S}}, \boldsymbol{y}_i \sim \mathcal{P}_Y^{\mathcal{S}}\}_{i=1}^{N_{\mathcal{S}}}$ at the beginning of CDSL. Here $\mathcal{P}_X$ and $\mathcal{P}_Y$ are the input and label distributions, respectively, while $N_{\mathcal{S}}$ is the sample quantity. This paper chooses visual recognition as the learning task, in which the number of data classes is $K$. Then, we consider a sequence of continually arriving target domains $\mathbf{T} = \{\mathcal{T}^t\}_{t=1}^T$, where $T$ denotes the total number of domains. The $t$-th domain contains $N_{\mathcal{T}^t}$ data samples $\boldsymbol{x}$ but no labels $\boldsymbol{y}$, i.e., $\mathcal{T}^t = \{\boldsymbol{x}_i \| \boldsymbol{x}_i \sim \mathcal{P}_X^{\mathcal{T}^t}\}_{i=1}^{N_{\mathcal{T}^t}}$, and all the target domains share the same label space with the source domain. Note that the domain order is randomly determined (including the source domain), which means that the superscript $t$ does not always indicate the same domain. The generalization performance for the $t$-th domain $\mathcal{T}^t$ depends on both the previously seen $t-1$ target domains and the original source domain, i.e., $\mathbf{S} = \{\mathcal{S}, \cup_{j=1}^{t-1} \mathcal{T}^j\}$. Considering the possible limitation on memory storage and regulations on privacy (Guo et al., 2021) or intellectual property (Wang et al., 2021b), like regular continual learning (Dong et al., 2022a), we also set an exemplar memory $\mathcal{M}$ to store $\frac{|\mathcal{M}|}{t}$ exemplars that are closest to the class centroids for each domain in $\mathbf{S}$, where $\frac{|\mathcal{M}|}{t} \ll N_{\mathcal{T}}$, $\frac{|\mathcal{M}|}{t} \ll N_{\mathcal{S}}$. Moreover, the memory size is set to be fixed during CDSL. We assume that the model is a deep neural network, and without loss of generality, the neural network consists of a feature extractor $f_\theta$ at the bottom and a classifier $g_\omega$ at the top. For each target domain dataset $\mathcal{T}_t$, in addition to the current model $f_\theta^t \circ g_\omega^t$, its inherited version $f_\theta^{t-1} \circ g_\omega^{t-1}$ from the last

domain is also stored for later usage. We aim to improve the generalization performance on a new target domain (TDG) before and during its training, i.e., in its *Unfamiliar Period* of CDSL, while also achieving effective target domain adaptation (TDA) and forgetting alleviation (FA).

## 2.2 RANDOM MIXUP AUGMENTATION

At the beginning of CDSL, only a labeled source domain is available. Thus we face a pure singe-domain generalization problem where the model is trained on a single domain $\mathcal{S}$ and tested on the other unseen domains $\mathbf{T}$ (each domain in $\mathbf{T}$ is possible as the domain order is random). We apply data augmentation and design a Random Mixup (RandMix) method to solve this problem. RandMix is not only helpful to improve cross-domain transferability from $\mathcal{S}$ to $\mathcal{T}^1$, but also beneficial for remaining model's order agnostic generalization when it encounters low-quality domains. RandMix relies on $N_{\text{aug}}$ simple autoencoders $\mathbf{R} = \{R_i\}_{i=1}^{N_{\text{aug}}}$, where each autoencoder consists of an encoder $e_\xi$ and a decoder $d_\zeta$. We want RandMix to be as simple as possible, preferably to work without training. Inspired by (Wang et al., 2021c), the encoder $e_\xi$ and the decoder $d_\zeta$ are implemented as a convolutional layer and a transposed convolutional layer. With such implementation, even if the parameters $\xi$ and $\zeta$ are randomly initialized from a normalized distribution, the autoencoder can still generate reasonable augmentation data. In order to introduce more randomness, we apply AdaIN (Karras et al., 2019) to inject noise into the autoencoder. Specifically, the used AdaIN contains two linear layers $l_{\phi_1}$, $l_{\phi_2}$, and when $l_{\phi_1}$ and $l_{\phi_2}$ are fed with a certain noisy input drawn from a normalized distribution ($n \sim \mathcal{P}_{\text{N}(0,1)}$), they can produce two corresponding noisy outputs with smaller variances. As observed in our experiments (Section 3.1), blindly pushing augmentation data away from the original (Li et al., 2021; Wang et al., 2021c) possibly hurts the model generalization ability, and injecting randomness with a smaller variance is a better solution. The two noisy outputs are injected to the representations of the autoencoder as multiplicative and additive noises:

$$R(\boldsymbol{x}) = d_\zeta \left( l_{\phi_1}(n) \times l_{\text{IN}}(e_\xi(\boldsymbol{x})) + l_{\phi_2}(n) \right), \tag{1}$$

where $l_{\text{IN}}(\cdot)$ represents an element-wise normalization layer. RandMix works as feeding training data to all autoencoders and mixing the outputs with random weights drawn from a normalized distribution ($\mathbf{w} = \{\text{w}_i \| \text{w}_i \sim \mathcal{P}_{\text{N}(0,1)}\}_{i=0}^{N_{\text{aug}}}$). Finally, the mixture is scaled by a sigmoid function $\sigma(x) = 1/(1 + e^{-x})$:

$$\mathbf{R}(\boldsymbol{x}) = \sigma \left( \frac{1}{\sum_{i=0}^{N_{\text{aug}}} \text{w}_i} \left[ \text{w}_0 \boldsymbol{x} + \sum_{i=1}^{N_{\text{aug}}} \left( \text{w}_i R_i(\boldsymbol{x}) \right) \right] \right). \tag{2}$$

In mini-batch training, every time there is a new data batch, parameters of autoencoders ($\xi$, $\zeta$), AdaIN-s ($\phi_1$, $\phi_2$), AdaIN noisy inputs ($n$) and mixup weights ($\mathbf{w}$) are all initialized again. With RandMix, we can generate augmentation data with the same labels corresponding to all labeled samples from the source domain. Then, these labeled augmentation samples will work together with the original source data and be fed into the model for training. However, for the following continually arriving target domains, conducting RandMix on all target data is unreasonable. In CDSL, all target domains $\mathbf{T} = \{\mathcal{T}^t\}_{t=1}^T$ are unlabeled. While we can apply approaches to produce pseudo labels, the supervision is likely unreliable and inaccurate. Therefore, to avoid error propagation and accumulation, we augment a subset of the target data rather than the full set. Specifically, we determine whether a target sample is supposed to be augmented based on its prediction confidence:

$$\tilde{\boldsymbol{x}} = \begin{cases} \mathbf{R}(\boldsymbol{x}), & \text{if} \max \left[ g_\omega(f_\theta(\boldsymbol{x})) \right]_K \geq r_{\text{con}} \\ \emptyset, & \text{otherwise} \end{cases}, \boldsymbol{x} \sim \mathcal{P}_X^{\mathcal{T}}, \tag{3}$$

where $\max[\cdot]_K$ denotes the maximum of a vector with $K$ dimensions, and $r_{\text{con}}$ is a confidence threshold that is set to 0.8 (sensitivity analysis of $r_{\text{con}}$ is provided in Appendix).

## 2.3 TOP$^2$ PSEUDO LABELING

In CDSL, all target domains arrive with only the data but no label. Thus we need a pseudo labeling mechanism to provide reasonably accurate supervision for subsequent optimizations. However, existing pseudo labeling approaches have various limitations. For example, softmax-based pseudo labeling (Lee et al., 2013) produces hard labels for unlabeled samples, but training with such hard labels sacrifices inter-class distinguishability within the softmax predictions. To preserve inter-class distinguishability, SHOT (Liang et al., 2020) proposes a clustering-based approach to align unlabeled samples with cluster centroids. However, when the model is applied in a new domain, it

is difficult to distinguish different classes since they are nearly overlapped together and the class boundary among them is extremely vague. Besides, such clustering-based methods usually use all training samples to construct centroids for each class. During clustering, for a particular cluster, samples that exactly belong to the same class are often not used well and those misclassified pose a negative impact on centroid construction. In this case, such negative impact hurts the pseudo labeling accuracy by resulting in low-quality centroids. To address these issues, we propose a novel mechanism called Top$^2$ Pseudo Labeling (T2PL). Specifically, we use the softmax confidence to measure the possibility of correct classification for data samples, and select the top 50% set to construct class centroids:

$$\mathcal{I}_k = \cup^{\frac{N_{\mathcal{T}^t}}{r_{\text{top}} \cdot K}} \left\{ \arg\max_{\boldsymbol{x} \in \mathcal{T}^t} \left[ g_{\omega,k}(f_\theta(\boldsymbol{x})) \right] \right\}, \mathcal{F} = \cup_{k=1}^K \left\{ \cup_{i \in \mathcal{I}_k, \boldsymbol{x} \in \mathcal{T}^t} \{\boldsymbol{x}_i\} \right\}, \tag{4}$$

where $g_{\omega,k}(\cdot)$ denotes the $k$-th element of the classifier outputs. $r_{\text{top}}$ controls the size of the selected top set, and if the data is class balanced, top 50% corresponds to $r_{\text{top}} = 2$. Then the top set $\mathcal{F}$ is used to construct class centroids by a prediction-weighted aggregation on representations:

$$p_k = \frac{\sum_{\boldsymbol{x}_i \in \mathcal{F}} g_{\omega,k}(f_\theta(\boldsymbol{x}_i)) \cdot f_\theta(\boldsymbol{x}_i)}{\sum_{\boldsymbol{x}_i \in \mathcal{F}} g_{\omega,k}(f_\theta(\boldsymbol{x}_i))}. \tag{5}$$

Subsequently, we can compute the cosine similarity between these centroids and representations of all current unlabeled data, and also select the top half set of samples that have much higher similarity to the centroids than the rest for each class:

$$\mathcal{I}_k' = \cup^{\frac{N_{\mathcal{T}^t}}{r_{\text{top}} \cdot K}} \left\{ \arg\max_{\boldsymbol{x} \in \mathcal{T}^t} \left[ \frac{f_\theta(\boldsymbol{x}) \cdot p_k^\top}{\|f_\theta(\boldsymbol{x})\|\|p_k\|} \right] \right\}, \mathcal{F}' = \cup_{k=1}^K \left\{ \cup_{i \in \mathcal{I}_k', \boldsymbol{x} \in \mathcal{T}^t} \{(\boldsymbol{x}_i, k)\} \right\}. \tag{6}$$

Then we use $\mathcal{F}'$ to fit a k-Nearest Neighbor (kNN) classifier and assign the pseudo label as below. Note that kNN can decentralize the risk of misclassification in assigning pseudo labels by a single comparison between class centroids and data samples, and is more suitable for clustering samples that are highly overlapped than center-based approaches:

$$\hat{\boldsymbol{y}} = \text{kNN}\left(\boldsymbol{x}, \mathcal{F}'\right)_{\text{Euclidean}}^{\frac{N_{\mathcal{T}^t}}{r_{\text{top}}' \cdot K}}, \tag{7}$$

where the superscript $\frac{N_{\mathcal{T}^t}}{r_{\text{top}}' \cdot K}$ and subscript Euclidean mean that the kNN works to find $\frac{N_{\mathcal{T}^t}}{r_{\text{top}}' \cdot K}$ closest neighbors from $\mathcal{F}'$ with the measurement of Euclidean distance. We select the top 5% for each class ($r_{\text{top}}' = 20$) in our implementation (sensitivity analyses of $r_{\text{top}}$ and $r_{\text{top}}'$ are provided in the Appendix). With this kNN, T2PL makes better use of unlabeled samples with a relatively high possibility of correct classification than SHOT, and provides more accurate pseudo labels.

## 2.4 PROTOTYPE CONTRASTIVE ALIGNMENT

Prototype learning (PL) (Pan et al., 2019; Kang et al., 2019; Tanwisuth et al., 2021; Dubey et al., 2021) has been demonstrated to be effective for solving cross-domain sample variability and sample insufficiency (Wang et al., 2021a). These two issues are reflected in our problem as the random uncertainty of augmentation data and limited data quantity of seen domains in the exemplar memory $\mathcal{M}$. As a result, in this work, we adopt the idea of PL and propose a new Prototype Contrastive Alignment (PCA) algorithm to align domains together for improving model generalization ability.

First, we need to point out that regular PL is unsuitable to our problem. In regular PL, all samples are fed into the model to extract representations, and these representations are aggregated class-by-class for constructing prototypes $\mathbf{p}^t = \{p_k^t\}_{k=1}^K$ of a certain domain $\mathcal{T}^t$. To achieve cross-domain alignment, prototypes of different domains for the same classes are aggregated and averaged. Then data samples from different domains are optimized to approach these domain-average prototypes. However, such domain alignment has a problem of adaptivity gap (Dubey et al., 2021). As shown in Figure 2, we assume that there is an optimal prototype location $p_k^*$ of class $k$ for all domains ($\mathcal{S} \cup \mathbf{T}$), and the objective is to move the source prototype $p_k^s$ to the optimal by training on target domains ($\mathbf{T}$) one-by-one. Meanwhile, we also regard that there are optimal prototypes $p_k^t$ for each domain itself, and suppose that the regular PL is applied to align prototypes of all these domains. To illustrate the problem of adaptivity gap, we take the alignment between the source $\mathcal{S}$ and the first target domain $\mathcal{T}^1$ as an example, and assume that the distance between $p_k^1$ and $p_k^*$ is larger than the distance between $p_k^s$ and $p_k^*$. In this case, after the domain alignment, the location of the current prototype

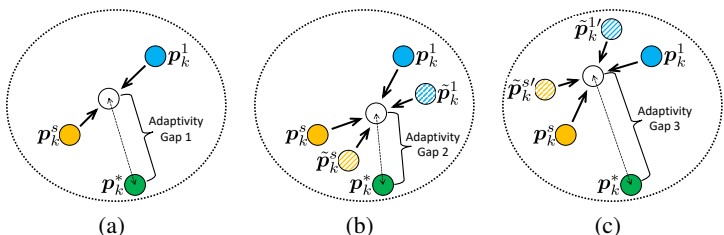 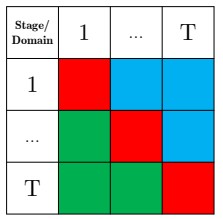

Figure 2: (a): When the model encounters a domain whose prototypes ($\boldsymbol{p}_k^1$) are far from the optimal ones ($\boldsymbol{p}_k^*$), domain alignment with regular prototype learning will enlarge the adaptivity gap. (b): `RandMix` may be helpful to reduce the gap in many cases of ($\tilde{\boldsymbol{p}}_k^s, \tilde{\boldsymbol{p}}_k^1$). (c): However, in some cases `RandMix` may produce low-quality data augmentation ($\tilde{\boldsymbol{p}}_k^{s\prime}, \tilde{\boldsymbol{p}}_k^{1\prime}$) that negatively affects the regular prototype learning and enlarges the gap. Here, we have Adaptivity Gap 3 > Adaptivity Gap 1 > Adaptivity Gap 2.

Figure 3: Accuracy Matrix. For each domain, TDG: mean of green elements; TDA: red elements; FA: mean of blue elements.

is worse than the source prototype, which means that the adaptivity gap is enlarged. Although we can use `RandMix` to generate more data ($\tilde{\boldsymbol{p}}_k^s, \tilde{\boldsymbol{p}}_k^1$) that might be helpful for reducing the adaptivity gap, there is randomness for each use of `RandMix` and we cannot guarantee such reduction every time (e.g., when the augmentation data is as $\tilde{\boldsymbol{p}}_k^{s\prime}$ and $\tilde{\boldsymbol{p}}_k^{1\prime}$ in Figure 2). Therefore, a better prototype construction strategy is needed to provide more representative prototypes.

Inspired by a semi-supervised learning work (Saito et al., 2019), we view neuron weights of the classifier $\omega$ as the prototypes. The classifier $g_\omega$ consists of a single linear layer and is placed exactly behind the feature extractor $f_\theta$. Thus the dimension of its neuron weights is the same as the hidden dimension of extracted representations. Prototypes $\boldsymbol{p}_k^t$-s are obtained with a CrossEntropy Loss:

$$\mathcal{L}_{\mathrm{CE}} = \mathbb{E}_{\boldsymbol{x}_i \sim \mathcal{P}_X^{\mathcal{T}^t}} \left[ \sum_{k=1}^K - \log \frac{\mathbb{I}_{\hat{\boldsymbol{y}}_i = k} \exp\left(\boldsymbol{p}_k^{t\top} f_\theta(\boldsymbol{x}_i) + b_k\right)}{\sum_{c=1}^K \exp\left(\boldsymbol{p}_c^{t\top} f_\theta(\boldsymbol{x}_i) + b_c\right)} \right], \tag{8}$$

where $b_k$, $b_c$ are biases of the linear layer, and we set them as zeros during training, and $\mathbb{I}_{(\cdot)}$ is an indicator function that if the subscript condition is true, $\mathbb{I}_{\mathrm{True}} = 1$, otherwise, $\mathbb{I}_{\mathrm{False}} = 0$.. Compared with regular prototypes, such linear layer prototypes are built on a lot more data because `RandMix` is initialized and used for every mini-batch during model training. Moreover, `RandMix` has general effect on improving generalization ability, as shown in our experiments later in Section 3.2. Thus we believe that linear layer prototypes have smaller adaptivity gaps than regular ones.

With linear layer prototypes, we can align domains in a better way. Unlike the alignment on regular prototypes, we do not sum up and average prototypes of different domains. Instead, we simultaneously maximize the similarities of data samples to prototypes of different domains, although in practice we may have prototypes of just two domains since there is only one old model being stored (Section 2.1). We also introduce contrastive comparison into the domain alignment process that can enhance the distinguishability of different classes and be beneficial for our pseudo labeling. The contrastive comparison includes negative pairs of a particular sample and samples from other classes. In this case, our prototype contrastive alignment is formulated as follows:

$$\mathcal{L}_{\mathrm{PCA}} = \mathbb{E}_{\boldsymbol{x}_i \sim \mathcal{P}_X^{\mathcal{T}^t}} \left\{ \sum_{k=1}^K - \log \frac{\mathbb{I}_{\hat{\boldsymbol{y}}_i = k} \left[\exp\left(\boldsymbol{p}_k^{t\top} f_\theta(\boldsymbol{x}_i)\right) + \exp\left(\boldsymbol{p}_k^{t-1\top} f_\theta(\boldsymbol{x}_i)\right)\right]}{\Delta} \right\}, \text{where}$$

$$\Delta = \sum_{c=1}^K \exp\left(\boldsymbol{p}_c^{t\top} f_\theta(\boldsymbol{x}_i)\right) + \sum_{c=1}^K \exp\left(\boldsymbol{p}_c^{t-1\top} f_\theta(\boldsymbol{x}_i)\right) + \sum_{\boldsymbol{x}_j \in \mathcal{T}^t, j \neq i} \mathbb{I}_{\hat{\boldsymbol{y}}_i \neq \hat{\boldsymbol{y}}_j} \exp\left(f_\theta(\boldsymbol{x}_i)^\top f_\theta(\boldsymbol{x}_j)\right). \tag{9}$$

Furthermore, we also adopt knowledge distillation from the stored old model to the current model for forgetting compensation. In this case, our final optimization objective is shown as follows:

$$\mathcal{L} = \mathcal{L}_{\mathrm{CE}} + \mathcal{L}_{\mathrm{PCA}} + \mathcal{L}_{\mathrm{DIS}}, \text{where } \mathcal{L}_{\mathrm{DIS}} = \mathcal{D}_{\mathrm{KL}} \left[ g_\omega^{t-1}\left(f_\theta^{t-1}(\boldsymbol{x})\right) \| g_\omega^t\left(f_\theta^t(\boldsymbol{x})\right) \right]_{\boldsymbol{x} \sim \mathcal{P}_X^{\mathcal{T}^t}}. \tag{10}$$

Note that this optimization objective is used in all target domains. As for the source domain, there is no distillation $\mathcal{L}_{\mathrm{DIS}}$ and the nominator of $\mathcal{L}_{\mathrm{PCA}}$ only contains the first term, because there is no stored old model in the stage of source training. We denote this simplified optimization loss as $\mathcal{L}'$.

## 3 EXPERIMENTS

Our code is available at https://github.com/SonyAI/RaTP. The datasets, experiment settings, and comparison baselines are introduced below, and more details are in the appendix.

**Datasets. Digits** consists of 5 different domains with 10 classes, including MNIST (MT), SVHN (SN), MNIST-M (MM), SYN-D (SD) and USPS (US). **PACS** contains 4 domains with 7 classes, including Photo (P), Art painting (A), Cartoon (C), and Sketch (S). **DomainNet** is the most challenging cross-domain dataset, including Quickdraw (Qu), Clipart (Cl), Painting (Pa), Infograph (In), Sketch (Sk) and Real (Re). Considering label noise and class imbalance, we follow Xie et al. (2022) to split a subset of DomainNet for our experiments.

**Experiment Settings.** We apply ResNet-50 as the feature extractor for both PACS and DomainNet, and apply DTN as the feature extractor for Digits (Liang et al., 2020). In the source domain, we randomly split 80% data as the training set and the rest 20% as the testing set. In target domains, all data are used for training and testing. The SGD optimizer with an initial learning rate of 0.01 is used for Digits, and 0.005 for PACS and DomainNet. The exemplar memory size is set as 200 for all datasets, and the batch size is 64. For all experiments, we conduct multiple runs with three seeds (2022, 2023, 2024), and report the average performance.

**Comparison Baselines.** RaTP is compared with a comprehensive set of state-of-the-art works from Continual DA [CoTTA (Wang et al., 2022), AuCID (Rostami, 2021)], Source-Free DA [SHOT (Liang et al., 2020), GSFDA (Yang et al., 2021), BMD (Qu et al., 2022)], Test-Time/Online DA [Tent (Wang et al., 2020), T3A (Iwasawa & Matsuo, 2021)], Single DG [L2D (Wang et al., 2021c), PDEN (Li et al., 2021)], Unified DA&DG [SNR (Jin et al., 2021)], and Multiple DG [PCL (Yao et al., 2022), EFDM (Zhang et al., 2022)].

Table 1: Performance comparisons between ours and other methods on Digits in TDG, TDA, and FA under two domain orders (shown with ↓). We blue and bold **the best**, and bold **the second best**.

| Order | | Metric | CoTTA | AuCID | SHOT | GSFDA | BMD | Tent | T3A | L2D | PDEN | SNR | PCL | EFDM | Ours |
|---|---|---|---|---|---|---|---|---|---|---|---|---|---|---|---|
| MT ↓ MM ↓ SN ↓ SD ↓ US | TDG | MM | 53.3 | 42.8 | 55.6 | 54.8 | 55.3 | 53.1 | 51.3 | **73.6** | 68.4 | 68.9 | 51.7 | 48.0 | **86.8** |
| | | SN | 23.6 | 16.1 | 25.9 | 28.2 | 24.9 | 29.7 | 22.9 | 35.4 | **35.6** | 35.4 | 22.8 | 12.6 | **43.7** |
| | | SD | 33.8 | 31.4 | 44.1 | 29.3 | 42.6 | 38.0 | 33.9 | 52.6 | 54.1 | **55.8** | 35.0 | 15.1 | **67.5** |
| | | US | **88.9** | 81.6 | 69.0 | 51.6 | 73.9 | 87.8 | 77.4 | 87.7 | 83.5 | 80.5 | 71.9 | 35.5 | **90.1** |
| | | Avg. | 49.9 | 43.0 | 48.7 | 41.0 | 49.2 | 52.2 | 46.4 | **62.3** | 60.4 | 60.2 | 45.4 | 27.8 | **72.0** |
| | TDA | MT | 99.0 | 99.1 | **99.4** | **99.4** | **99.4** | 99.0 | 99.0 | **99.3** | 99.1 | 99.2 | 99.2 | 99.2 | **99.4** |
| | | MM | 40.7 | 42.3 | 85.3 | 54.5 | 80.7 | 52.8 | 50.2 | **86.1** | 84.7 | 84.1 | 53.0 | 12.9 | **90.2** |
| | | SN | 18.6 | 18.2 | 33.4 | 13.1 | 34.7 | 25.5 | 23.2 | **53.8** | 50.8 | 53.3 | 22.6 | 8.0 | **69.1** |
| | | SD | 25.4 | 30.0 | 41.6 | 13.1 | 45.4 | 37.2 | 31.2 | **58.7** | 51.6 | 43.7 | 28.2 | 10.5 | **81.8** |
| | | US | 89.5 | 79.6 | 75.7 | 14.7 | 81.2 | 85.1 | 83.5 | **90.3** | 90.2 | 90.2 | 58.0 | 15.4 | **93.4** |
| | | Avg. | 54.6 | 53.8 | 67.1 | 39.0 | 68.3 | 59.9 | 57.4 | **77.6** | 75.3 | 74.1 | 52.2 | 29.2 | **86.8** |
| | FA | MT | **93.6** | **93.6** | 73.6 | 33.9 | 78.8 | 93.2 | 88.4 | 92.2 | 91.2 | 91.9 | 83.0 | 31.6 | **95.4** |
| | | MM | 32.9 | 49.2 | 39.0 | 10.0 | 44.2 | 50.1 | 46.4 | 70.8 | 69.7 | **70.9** | 36.3 | 15.4 | **82.1** |
| | | SN | 16.5 | 22.5 | 16.0 | 8.5 | 17.3 | 24.7 | 16.0 | **56.0** | 50.4 | 54.2 | 20.6 | 10.3 | **70.5** |
| | | SD | 28.4 | 30.0 | 41.1 | 10.8 | 45.7 | 35.3 | 26.1 | 70.1 | 68.8 | **71.1** | 27.2 | 15.7 | **81.3** |
| | | Avg. | 42.9 | 48.8 | 42.4 | 15.8 | 46.5 | 50.8 | 44.2 | **72.3** | 70.0 | 72.0 | 41.8 | 18.3 | **82.3** |
| US ↓ SD ↓ SN ↓ MM ↓ MT | TDG | SD | 36.6 | 35.8 | 37.7 | 37.2 | 37.4 | 37.5 | 36.4 | **61.3** | 60.2 | 60.7 | 35.3 | 37.0 | **68.6** |
| | | SN | 17.6 | 19.2 | 18.4 | 22.0 | 18.3 | 27.2 | 20.6 | **52.7** | 51.5 | 52.2 | 21.4 | 13.4 | **62.0** |
| | | MM | 29.7 | 36.2 | 25.3 | 33.4 | 29.4 | 42.1 | 34.0 | **64.5** | 60.8 | 60.2 | 33.5 | 17.7 | **68.7** |
| | | MT | 52.8 | 63.4 | 37.4 | 55.6 | 56.5 | 76.7 | 61.0 | **83.4** | 82.7 | 80.7 | 62.0 | 29.0 | **87.3** |
| | | Avg. | 34.2 | 38.7 | 29.7 | 37.1 | 35.4 | 45.9 | 38.0 | **65.5** | 63.8 | 63.5 | 38.1 | 24.3 | **71.7** |
| | TDA | US | 98.5 | 98.6 | 98.6 | 98.3 | 98.6 | 98.5 | 98.5 | **99.0** | 98.8 | 98.9 | 98.6 | 98.3 | **98.9** |
| | | SD | 23.2 | 24.3 | 32.9 | 40.2 | 30.9 | 39.8 | 25.7 | **74.8** | 71.8 | 71.8 | 37.1 | 9.6 | **82.6** |
| | | SN | 13.2 | 17.6 | 21.6 | 17.9 | 23.4 | 29.5 | 22.0 | **72.1** | 67.4 | 64.7 | 19.5 | 10.1 | **78.7** |
| | | MM | 25.7 | 30.4 | 19.4 | 20.0 | 29.6 | 39.7 | 36.1 | **78.9** | 76.7 | 74.4 | 29.4 | 10.8 | **81.8** |
| | | MT | 42.8 | 53.4 | 30.3 | 19.9 | 53.8 | 78.8 | 64.3 | **89.6** | 86.8 | 83.2 | 54.8 | 19.8 | **93.3** |
| | | Avg. | 40.7 | 44.9 | 40.6 | 39.3 | 47.3 | 57.3 | 49.3 | **82.9** | 80.3 | 78.6 | 47.9 | 29.7 | **87.1** |
| | FA | US | 95.2 | 90.3 | 44.2 | 49.9 | 57.2 | **98.5** | 80.4 | 93.4 | 90.5 | 91.3 | 81.8 | 10.7 | **96.2** |
| | | SD | 20.1 | 25.7 | 22.3 | 22.4 | 26.1 | 37.7 | 25.9 | **71.9** | 70.7 | 68.8 | 26.6 | 11.0 | **82.9** |
| | | SN | 11.8 | 19.1 | 13.5 | 8.6 | 14.1 | 28.6 | 13.8 | **56.8** | 55.6 | 55.0 | 17.7 | 14.3 | **68.4** |
| | | MM | 26.1 | 29.1 | 14.5 | 10.1 | 22.7 | 41.2 | 14.9 | **74.4** | 71.3 | 70.9 | 29.1 | 27.2 | **83.5** |
| | | Avg. | 38.3 | 41.1 | 23.6 | 22.8 | 30.0 | 51.5 | 33.8 | **74.1** | 72.0 | 71.5 | 38.8 | 15.8 | **82.8** |

### 3.1 Effectiveness of the RaTP Framework

We assume that each domain corresponds to a single training stage. After the training of every stage, we test the model performance on all domains. In this case, we can obtain an accuracy matrix, as shown in Figure 3. Target domain generalization (TDG) is measured by the mean model performance on a domain *before* its training stage (mean of the green elements in Figure 3), target domain adaptation (TDA) by the performance on a domain *right after* finishing its training stage (the red element), and forgetting alleviation (FA) by the mean performance on a domain *after* the model has been updated with other domains (mean of the blue elements). All these metrics are the higher, the better. Due to space limitation, we report experiment results for Digits (Tables 1) and PACS (Table 2) in two domain orders, and for DomainNet (Table 3) in one domain order. Results for more domain orders can be found in the Appendix. We can observe that **RaTP achieves significantly higher performance in TDG than all baselines in all cases, and comparable (and in many cases higher) performance in TDA and FA.** Specifically, RaTP significantly outperforms the second-best baseline on TDG in all cases and by 3.1∼10.3% in average, and the performance improvement is particularly large over DA methods. For TDA and FA, RaTP is also the best in most cases (and clearly outperforms the second best in many cases); and if not, very close to the best. These results demonstrate that our method can effectively improve the model performance during the *Unfamiliar Period*, adapt the model to different domains, and alleviate catastrophic forgetting on seen domains.

Table 2: Performance comparisons between ours and other methods on PACS in TDG, TDA, and FA under two domain orders (shown with ↓). We blue and bold **the best**, and bold **the second best**.

| Order | Metric | | CoTTA | AuCID | SHOT | GSFDA | BMD | Tent | T3A | L2D | PDEN | SNR | PCL | EFDM | Ours |
|---|---|---|---|---|---|---|---|---|---|---|---|---|---|---|---|
| | TDG | A | 71.1 | 67.0 | 68.2 | **75.0** | 68.4 | 71.6 | 71.1 | 67.5 | 67.0 | 67.4 | 66.4 | 66.5 | **73.9** |
| | | C | 47.0 | 29.9 | 32.0 | 44.9 | 34.9 | 48.5 | 35.9 | **49.2** | 47.8 | 42.7 | 42.5 | 37.7 | **53.1** |
| | | S | 46.3 | 40.8 | 29.0 | 46.4 | 31.3 | 48.6 | 38.1 | **52.7** | 52.5 | 45.3 | 41.4 | 44.0 | **60.7** |
| P | | Avg. | 54.8 | 45.9 | 43.1 | 55.4 | 44.9 | 56.2 | 48.4 | **56.5** | 55.8 | 51.8 | 50.1 | 49.4 | **62.6** |
| ↓ | TDA | P | **99.4** | **99.4** | 97.0 | 98.8 | 97.0 | **99.4** | **99.4** | 99.1 | 99.2 | 99.0 | 98.8 | 99.1 | **99.5** |
| A | | A | 77.0 | 64.6 | 85.0 | 82.1 | **86.9** | 76.6 | 74.2 | 72.8 | 70.2 | 71.8 | 77.4 | 72.4 | **87.5** |
| ↓ | | C | 64.3 | 58.0 | 70.4 | **81.6** | 75.0 | 68.2 | 66.7 | 61.4 | 60.9 | 61.2 | 61.5 | 37.6 | **75.1** |
| C | | S | 54.4 | 54.3 | 62.9 | **81.5** | 61.3 | 58.7 | 57.1 | **68.6** | 65.8 | 65.4 | 65.6 | 30.0 | 67.1 |
| ↓ | | Avg. | 73.8 | 69.1 | 78.8 | **86.0** | 80.1 | 75.7 | 74.4 | 75.5 | 74.0 | 74.4 | 75.8 | 59.8 | **82.3** |
| S | FA | P | **99.4** | 96.6 | 94.9 | 71.0 | 94.3 | **99.4** | 95.9 | 91.1 | 93.2 | 96.9 | 77.1 | 79.3 | **97.9** |
| | | A | 77.1 | 74.7 | 79.1 | 42.0 | **79.8** | 76.2 | 61.9 | 64.4 | 64.1 | 64.6 | 60.2 | 56.8 | **87.8** |
| | | C | 64.5 | 64.6 | 66.3 | 23.1 | 60.9 | **68.7** | 55.7 | 57.6 | 55.5 | 60.2 | 40.1 | 46.1 | **73.0** |
| | | Avg. | 80.3 | 78.6 | 80.1 | 45.4 | 78.3 | **81.4** | 71.2 | 71.0 | 70.9 | 73.9 | 59.1 | 60.7 | **86.2** |
| | TDG | C | 58.4 | 50.5 | 46.4 | **70.1** | 46.5 | 58.3 | 58.4 | 63.7 | 63.1 | 63.1 | 50.1 | 64.2 | **69.6** |
| | | A | 49.8 | 40.8 | 42.0 | 46.7 | 43.7 | 50.4 | 47.8 | **52.0** | 50.8 | 51.5 | 36.4 | 33.6 | **68.7** |
| S | | P | 54.7 | 50.8 | **71.3** | 60.0 | 70.9 | 54.2 | 63.7 | 70.2 | 68.9 | 70.0 | 63.3 | 34.9 | **83.0** |
| ↓ | | Avg. | 54.3 | 47.4 | 53.2 | 58.9 | 53.7 | 54.3 | 56.6 | **62.0** | 60.9 | 61.5 | 49.9 | 44.2 | **73.8** |
| C | TDA | S | 95.4 | 95.5 | 94.7 | 96.2 | 94.7 | 95.4 | 95.4 | **96.3** | 94.9 | 96.0 | 96.1 | 95.9 | **96.5** |
| ↓ | | C | 68.3 | 65.3 | **86.6** | 79.4 | **87.9** | 67.1 | 75.3 | 79.5 | 79.7 | 80.0 | 73.9 | 70.1 | 77.1 |
| A | | A | 58.0 | 67.9 | **88.1** | 83.4 | **88.9** | 57.3 | 72.9 | 69.7 | 68.3 | 69.8 | 55.7 | 14.5 | 87.7 |
| ↓ | | P | 55.2 | 75.1 | 95.0 | 69.0 | **95.8** | 58.5 | 77.9 | 86.6 | 85.9 | 87.0 | 89.6 | 9.8 | **97.3** |
| P | | Avg. | 69.2 | 76.0 | **91.1** | 82.0 | **91.8** | 69.6 | 80.4 | 83.0 | 82.5 | 83.2 | 78.8 | 47.6 | 89.7 |
| | FA | S | **96.1** | 72.4 | 88.5 | 89.6 | 87.6 | **96.1** | 86.2 | 87.1 | 88.6 | 88.1 | 90.7 | 83.4 | **94.0** |
| | | C | 68.5 | 69.9 | **84.7** | 67.7 | **86.6** | 67.8 | 64.9 | 76.7 | 75.3 | 76.6 | 72.0 | 53.5 | 80.8 |
| | | A | 58.1 | 51.7 | 84.6 | 76.4 | **84.7** | 58.3 | 61.6 | 70.3 | 70.9 | 69.1 | 61.2 | 20.6 | **87.7** |
| | | Avg. | 74.2 | 64.7 | 85.9 | 77.9 | **86.3** | 74.1 | 70.9 | 78.0 | 78.3 | 77.9 | 74.7 | 52.5 | **87.5** |

### 3.2 Ablation Studies

To demonstrate the effectiveness of each module in RaTP, we conduct comprehensive ablation studies for RandMix, T2PL, and PCA on Digits with the domain order US→SD→SN→MM→MT.

**Effectiveness of RandMix.** We first remove RandMix from RaTP and see how the model performance will change. We also try to apply RandMix on a few other baselines, including SHOT (Liang et al., 2020), GSFDA (Yang et al., 2021), and PCL (Yao et al., 2022). The experiment results in Figure 4(a) demonstrate that, removing RandMix from RaTP hurts the performance significantly, and attaching RandMix to other baseline methods clearly improves the model performance. Such observations prove the effectiveness of RandMix.

Table 3: Performance comparisons between ours and other methods on DomainNet in TDG, TDA, and FA (domain order is shown left with ↓). We blue and bold the best, and bold **the second best**.

| Order | | Metric | CoTTA | AuCID | SHOT | GSFDA | BMD | Tent | T3A | L2D | PDEN | SNR | PCL | EFDM | Ours |
|---|---|---|---|---|---|---|---|---|---|---|---|---|---|---|---|
| | TDG | Sk | 29.4 | 27.9 | 22.7 | 29.1 | 23.1 | 29.4 | 29.4 | **31.6** | 29.6 | 30.6 | 27.1 | 31.4 | **33.4** |
| | | Cl | 47.6 | 42.4 | 22.2 | 52.0 | 25.0 | 50.3 | 45.4 | **52.3** | 49.4 | 51.4 | 41.1 | 27.3 | **54.6** |
| | | In | 16.0 | **16.7** | 13.0 | 14.0 | 11.8 | **16.6** | 14.8 | 13.2 | 12.2 | 9.7 | 12.7 | 12.6 | 16.2 |
| Qu | | Pa | **28.4** | 23.9 | 15.0 | 26.4 | 16.5 | 27.3 | 23.0 | 24.2 | 22.6 | 26.2 | 13.3 | 14.0 | **41.0** |
| ↓ | | Re | 52.8 | 41.2 | 30.0 | 41.1 | 33.9 | **53.2** | 47.7 | 39.8 | 37.3 | 35.8 | 35.9 | 20.4 | **62.2** |
| Sk | | Avg. | 34.8 | 30.4 | 20.6 | 32.5 | 22.1 | **35.4** | 32.1 | 32.2 | 30.2 | 30.7 | 26.0 | 21.1 | **41.5** |
| ↓ | TDA | Qu | **92.7** | **92.7** | 92.0 | **94.8** | 92.0 | 92.0 | 92.1 | 91.8 | 92.0 | 91.8 | 91.5 | 90.9 | 92.0 |
| Cl | | Sk | 37.9 | 31.1 | 34.9 | **46.5** | 37.2 | 38.2 | 36.9 | 37.1 | 34.2 | 35.0 | 34.1 | 12.2 | **49.8** |
| ↓ | | Cl | 56.7 | 54.0 | 34.9 | **59.0** | 35.7 | 57.2 | 56.2 | 55.2 | 52.3 | 56.2 | 48.0 | 8.5 | **64.8** |
| In | | In | 17.5 | **19.3** | 17.2 | **22.5** | 12.9 | 18.9 | 18.2 | 13.2 | 12.1 | 12.6 | 17.5 | 11.5 | 19.0 |
| ↓ | | Pa | **34.8** | 30.6 | 19.6 | 26.9 | 23.4 | 33.6 | 33.2 | 16.3 | 12.4 | 20.7 | 10.3 | 13.5 | **49.1** |
| Pa | | Re | 58.2 | 46.5 | 34.5 | 42.9 | 37.3 | **60.2** | 57.2 | 42.3 | 39.6 | 41.8 | 33.9 | 4.5 | **66.5** |
| ↓ | | Avg. | 49.6 | 45.7 | 38.9 | 48.8 | 39.8 | **50.0** | 49.0 | 42.7 | 40.4 | 43.0 | 39.2 | 23.5 | **56.8** |
| Re | FA | Qu | **91.9** | 85.7 | 42.1 | 69.7 | 33.5 | **91.3** | 73.6 | 79.8 | 76.5 | 81.9 | 80.6 | 64.0 | 86.1 |
| | | Sk | 38.6 | 34.8 | 29.0 | 33.8 | 29.2 | **39.3** | 33.1 | 32.7 | 31.4 | 33.0 | 29.0 | 13.8 | **51.3** |
| | | Cl | 56.6 | 53.0 | 29.2 | 36.8 | 32.2 | **58.3** | 48.6 | 44.5 | 43.2 | 42.9 | 41.8 | 21.7 | **63.8** |
| | | In | 17.9 | 17.5 | 15.4 | 12.3 | 12.8 | **19.3** | 17.6 | 13.1 | 11.8 | 10.9 | 14.0 | 12.5 | **19.6** |
| | | Pa | **35.1** | 28.9 | 21.5 | 33.3 | 22.3 | 34.4 | 33.7 | 15.8 | 16.9 | 23.7 | 5.6 | 5.0 | **47.8** |
| | | Avg. | 48.0 | 44.0 | 27.4 | 37.2 | 26.0 | **48.5** | 41.3 | 37.2 | 36.0 | 38.5 | 34.2 | 23.4 | **53.7** |

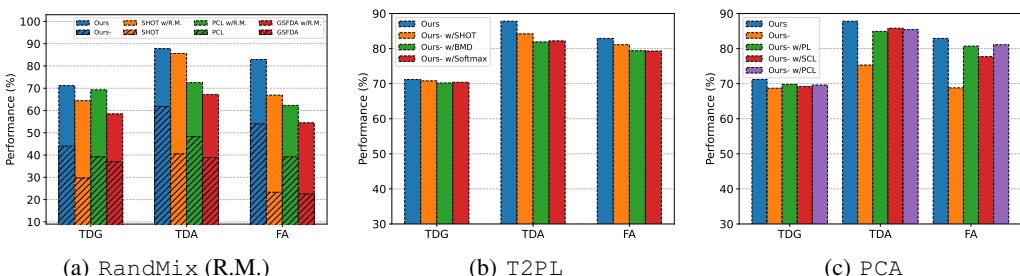

(a) RandMix (R.M.)  (b) T2PL  (c) PCA

Figure 4: Ablation studies of RandMix, T2PL, and PCA on Digits. The average performance of all domains for three metrics (TDG, TDA, FA) is shown. 'Ours' here denotes the full framework of RaTP, while 'Ours-' represents to remove a corresponding module (a, b or c) from RaTP, 'Ours-w/' means replacing the corresponding module with a new one.

**Effectiveness of T2PL.** We replace T2PL in RaTP with other pseudo labeling approaches to investigate the gain from T2PL, including Softmax (Lee et al., 2013), SHOT (Liang et al., 2020), and BMD (Qu et al., 2022). Figure 4(b) shows that RaTP performs the best when it has T2PL.

**Effectiveness of PCA.** We first remove $\mathcal{L}_{PCA}$ from the optimization objective, leaving only CrossEntropy Loss and Logits Distillation. We then try to replace PCA with regular forwarding prototype learning (PL), supervised contrastive learning (SCL), and PCL (Yao et al., 2022). Figure 4(c) shows that the performance of RaTP degrades significantly without $\mathcal{L}_{PCA}$, and when we use other methods instead, the performance is still worse. This validates the effectiveness of PCA.

## 4 CONCLUSION

This paper focuses on improving the model performance *before and during* the training of continually arriving domains, in what we call the *Unfamiliar Period*. To solve this practical and important problem, we propose a novel framework RaTP that includes a training-free data augmentation module RandMix, a pseudo labeling mechanism T2PL, and a prototype contrastive alignment training algorithm PCA. Extensive experiments demonstrate that RaTP can significantly improve the model performance in the *Unfamiliar Period*, while also achieving good performance in target domain adaptation and forgetting alleviation, in comparison with state-of-the-art methods from Continual DA, Source-Free DA, Test-Time/Online DA, Single DG, Multiple DG, and Unified DA&DG.

## ACKNOWLEDGEMENT

We gratefully acknowledge the support by Sony AI, National Science Foundation awards #2016240, #1834701, #1724341, #2038853, Office of Naval Research grant N00014-19-1-2496, and research awards from Meta and Google.

## ETHICS STATEMENT

In this paper, our studies are not related to human subjects, practices to data set releases, discrimination/bias/fairness concerns, and also do not have legal compliance or research integrity issues. Our work is proposed to address continual domain shifts when applying deep learning models in real-world applications. In this case, if the trained models are used responsibly for good purposes, we believe that our proposed methods will not cause ethical issues or pose negative societal impacts.

## REPRODUCIBILITY STATEMENT

The source code is provided at https://github.com/SonyAI/RaTP. All datasets we use are public. In addition, we also provide detailed experiment parameters and random seeds in the Appendix.

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

# SUMMARY OF THE APPENDIX

This Appendix includes additional details for the ICLR 2023 paper *"Deja Vu: Continual Model Generalization for Unseen Domains"*, including related work comparison, more implementation details of main experiments, additional experiment results, and sensitivity analysis on parameters used in the framework. The Appendix is organized as follows:

- Section A discusses the difference between `RaTP` and related works.

- Section B provides more implementation details, including detailed experiment settings, data splitting strategy, implementation details of the baseline methods, and network structures.

- Section C presents experiment results of additional domain orders on three datasets.

- Section D provides detailed sensitivity analysis of three parameters used in `RaTP`.

- Section E shows additional evaluations of `RaTP` in stationary domain adaptation.

## A    RELATED WORK COMPARISON

Our work is relevant to a number of different topics as we focus on addressing important and practical problems. We show a detailed comparison among these relevant topics in Table 4. Specifically, Fine-Tuning (Tajbakhsh et al., 2016) is usually conducted on a trained model with labeled target domain data. It can achieve certain domain adaptation capabilities after sufficient training, but the forgetting of previously learned knowledge is often severe. Continual Learning (Lopez-Paz & Ranzato, 2017; Rebuffi et al., 2017), which shares the same settings as Fine-Tuning, can effectively achieve alleviation of catastrophic forgetting. To relax the demand for target labels, a series of domain adaptation (DA) methods are proposed. Unsupervised DA (Ganin & Lempitsky, 2015; Dong et al., 2021; Zhu et al., 2021; Xu et al., 2021) can work effectively when the target data is unlabeled, but suffers from poor forgetting alleviation capability in continually changing cases. Continual DA (Wang et al., 2022; Rostami, 2021; Tang et al., 2021) is able to compensate for the forgetting of learned knowledge, while achieving good domain adaptation, but it requires access to source data. Source-Free DA (Yang et al., 2021; Liang et al., 2020; 2021; Qu et al., 2022) is proposed to adapt the model to unlabeled target domains without the need for source data, but the model forgets a lot of learned knowledge during adaptation. Test-Time DA (Wang et al., 2020; Iwasawa & Matsuo, 2021; Chen et al., 2022; Niu et al., 2022; Hong et al., 2023) does not need source data and target labels, and can achieve effective domain adaptation without heavy catastrophic forgetting on old knowledge. However, all these DA methods cannot achieve effective target domain generalization on new domains *before and during* the training on them, which is the main focus of our work here.

Different from DA, Domain Generalization (DG), as another cross-domain learning paradigm, can improve model out-of-distribution generalization (Dong et al., 2022b; Hong et al., 2021) performance even without seeing target domains. According to the composition of the given source domain, DG can be divided into two types, Single Source (Wang et al., 2021c; Li et al., 2021; Zhu & Li, 2022) and Multi-Source (Yao et al., 2022; Zhang et al., 2022). Both these two DG-s require labeled source domain data, and can address target domain generalization before any adaptation on target domains. However, they cannot achieve effective target domain adaptation and forgetting alleviation, which are important in continually changing scenarios. Recently, there are some works (Jin et al., 2021) that try to unify DA and DG, but they require significant resources such as source-labeled data and target-labeled data. Moreover, such Unified DA & DG also suffers from forgetting when being applied in continual learning settings. Compared with these studies, our work focuses on improving model performance on new target domains *before and during* the training on them, in what we call the *Unfamiliar Period* in the main text, while also achieving effective target domain adaptation and forgetting alleviation. To ensure the application practicability, we assume that there is only unlabeled data from target domains. As shown in the main text, the proposed `RaTP` framework can effectively solve the problem we target under such assumptions.

## B    IMPLEMENTATION DETAILS

**Experiment Settings.** To classify the datasets, we apply ResNet-50 as the feature extractor for both PACS and DomainNet. For Digits, we follow Liang et al. (2020) to use DTN as the feature extractor. In source domain, we follow Wang & Lu to split 80% data as the training set and the rest 20% as the testing set. In all target domains, all data are used for training and testing. For each dataset,

Table 4: Characteristics of various problem settings that adapt a model to potentially shifted target domains, especially assuming the domain shift is continually changing.

| Topics | Source Data | Target Data | Target Label | Target Domain Generalization | Target Domain Adaptation | Forgetting Alleviation |
|---|---|---|---|---|---|---|
| Fine-Tuning | ✗ | ✓ | ✓ | ✗ | ✓ | ✗ |
| Continual Learning | ✗ | ✓ | ✓ | ✗ | ✓ | ✓ |
| Unsupervised DA | ✓ | ✓ | ✗ | ✗ | ✓ | ✗ |
| Continual DA | ✓ | ✓ | ✗ | ✗ | ✓ | ✓ |
| Source-Free DA | ✗ | ✓ | ✗ | ✗ | ✓ | ✗ |
| Test-Time DA | ✗ | ✓ | ✗ | ✗ | ✓ | ✓ |
| Single Source DG | ✓ | ✗ | ✗ | ✓ | ✗ | ✗ |
| Multi-Source DG | ✓ | ✗ | ✗ | ✓ | ✗ | ✗ |
| Unified DA & DG | ✓ | ✓ | ✓ | ✓ | ✓ | ✗ |
| RaTP | ✗ | ✓ | ✗ | ✓ | ✓ | ✓ |

we keep the training steps per epoch the same for all domains in all different domain orders. We use 800 steps for Digits, 50 steps for PACS and 70 steps for DomainNet. Training epoch is 30 for all datasets and domains. The SGD optimizer with initial learning rate 0.01 is used for Digits and 0.005 for PACS and DomainNet. Moreover, we use momentum of 0.9 and weight decay of 0.0005 to schedule the SGD. The exemplar memory is set as 200 for all datasets, and we set the batch size of mini-batch training as 64. Only ResNet-50 is initialized as the pre-trained version of ImageNet.

**Subset of DomainNet.** The original DomainNet is class imbalanced, with certain classes in some domains containing very few images ($\sim$10). This makes it hard to assign pseudo labels for these classes. Thus, we refer to Xie et al. (2022) to select the top 10 classes with most images in all domains, which contain 26,013 samples in total. This dataset is still class imbalanced, with the smallest sample number for a class is 32, while the largest is 901. Therefore, this subset is still quite challenging.

**Baseline Implementations.** For fair comparison, we try our best to extend all baseline methods in our problem settings. AuCID (Rostami, 2021) shares the most similar setting with us. But their exemplar memory size is not limited and can be enlarged as more domains appear. And they transform Digits to gray scale images while we treat them as RGB images. For Test-Time/Online DA [CoTTA (Wang et al., 2022), Tent (Wang et al., 2020), T3A Iwasawa & Matsuo (2021), Ada-Con (Chen et al., 2022), EATA (Niu et al., 2022)], we keep the setting that adapts the model on each target domain only once. For Single DG [L2D (Wang et al., 2021c), PDEN (Li et al., 2021)] and Multiple DG [PCL (Yao et al., 2022), EFDM (Zhang et al., 2022)], we use SHOT (Liang et al., 2020) to assign pseudo labels for the optimization on target domains. In EFDM, we use samples from current domain as the content images and randomly select images in the replay memory as the style images. SNR (Jin et al., 2021) directly modifies the model structures to match statistics of different domains, thus it can be viewed as a unified approach for both DA and DG. In our implementation, we use SNR to modify the convolution filters of the feature extractor and optimize the model with SHOT. We also consider a more advanced version of SHOT for further comparison, i.e., SHOT++ (Liang et al., 2021), which equips an additional semi-supervised training step using Mix-Match (Berthelot et al., 2019). This may not be fair for other methods because all methods can in principle equip with this additional training step. Thus, we also run SHOT++ without the MixMatch and denote it as SHOT+. All baseline methods are equipped with the same exemplar memory and the same feature extractor.

**Network Structures.** Our `RandMix` augmentation network includes four autoencoders that consist of a convolutional layer and a transposed convolutional layer. All these layers have 3 channels and kernel sizes of 5, 9, 13, 17, respectively. The classification model contains a feature extractor, a bottleneck module and a classifier. Specifically, pre-trained ResNet-50 is used as the feature extractor for both PACS and DomainNet. Following Liang et al. (2020), DTN, a variant of LeNet is used as the feature extractor for Digits. We also introduce a bottleneck module between the feature extractor and the classifier. The bottleneck consists of a fully-connected layer (256 units for Digits and 512 units for both PACS and DomainNet), a Batch Normalization layer, a ReLU layer and another fully-connected layer (128 units for Digits, 256 units for PACS, and 512 units

for DomainNet). The classifier is a fully-connected layer without the parameters of bias. Both contrastive loss and distillation loss are applied to the representations after the bottleneck.

## C ADDITIONAL EXPERIMENT RESULTS

Following existing studies of continual domain shifts (Rostami, 2021; Wang et al., 2022; Tang et al., 2021), we have presented experiment results under the most popular domain orders in the main text. To further explore the impact of domain order, which is rarely studied by existing works, we conduct more experiments with additional domain orders for the three datasets. Specifically, for each dataset, we carry out experiments under 10 different domain orders that are randomly shuffled, shown in Table 5. We present the average performance for all orders in Table 6 (Digits), Table 7 (PACS), Table 8 (DomainNet). Note that for the average performance in Tables 6, 7 and 8, we select a subset of the baseline methods that perform relatively better than the rest, including a few additional baselines (i.e., SHOT+ (Liang et al., 2021), SHOT++ (Liang et al., 2021), AdaCon (Chen et al., 2022), EATA (Niu et al., 2022), L2D (Wang et al., 2021c), and PDEN (Li et al., 2021)), to carry out experiments for saving time and space. Under these additional domain orders, `RaTP` still achieves the best average TDG on all three datasets, and comparable (and in many cases better) performance in TDA and FA. Combining these results with the ones shown in the main text, we can clearly see that `RaTP` provides significant improvement on TDG over previous state-of-the-art methods, while achieving comparable performance in TDA and FA.

Table 5: Different domain orders used to assess their impact on various approaches' performance. The results are reported in Tables 6, 7, and 8 for the average performance of all orders.

| Order Datasets | Digits | PACS | DomainNet |
|---|---|---|---|
| Order 1 | SN→MT→MM→SD→US | A→C→P→S | Re→Pa→In→Cl→Sk→Qu |
| Order 2 | SN→SD→MT→US→MM | A→C→S→P | Cl→In→Pa→Qu→Re→Sk |
| Order 3 | MM→US→MT→SD→SN | A→P→C→S | Cl→Re→In→Qu→Sk→Pa |
| Order 4 | MT→MM→SN→SD→US | C→A→S→P | In→Qu→Cl→Pa→Re→Sk |
| Order 5 | MT→MM→US→SN→SD | C→S→P→A | Pa→Sk→Qu→In→Re→Cl |
| Order 6 | SD→MM→SN→MT→US | P→A→C→S | Qu→Re→Cl→Pa→In→Sk |
| Order 7 | SD→SN→US→MM→MT | P→S→A→C | Qu→Sk→Cl→In→Pa→Re |
| Order 8 | SD→US→MM→SN→MT | P→S→C→A | Sk→In→Pa→Cl→Re→Qu |
| Order 9 | US→MT→SN→MM→SD | S→C→A→P | Sk→Re→Pa→Cl→Qu→In |
| Order 10 | US→SD→SN→MM→MT | S→P→C→A | Sk→Re→Qu→Pa→In→Cl |

## D SENSITIVITY ANALYSIS

We conduct sensitive analysis of the confidence threshold $r_{con}$ in splitting `RandMix`, and $r_{\text{top}}$ and $r'_{\text{top}}$ in `T2PL`. All sensitive analysis experiments are conducted on Digits with the domain order of US→SD→SN→MM→MT. For splitting `RandMix`, $r_{\text{con}}$ controls the proportion of unlabeled data that needs to be augmented. As shown in Figure 5(a), different $r_{\text{con}}$-s correspond to different model performance, and we can obtain the best performance when $r_{\text{con}} = 0.8$, which is adopted for Digits and DomainNet. And $r_{\text{con}}$ is set as 0.5 for PACS. As for the sensitivity analysis of the other two parameters, according to the results shown in Figures 5(b) and 5(c), we can observe that a larger $r_{\text{top}}$ that corresponds to a smaller amount of fitting samples is detrimental to the model performance, while a larger $r'_{\text{top}}$ that corresponds to fewer nearest samples for the kNN classification leads to slightly better performance. We choose $r_{\text{top}} = 2$ and $r'_{\text{top}} = 20$ for Digits and PACS, $r_{\text{top}} = 1$ and $r'_{\text{top}} = 10$ for DomainNet.

## E EVALUATIONS ON STATIONARY DOMAIN ADAPTATION

While our work primarily focuses on continual domain shift, here we provide results of stationary domain adaptation for further comparison. In this setting, we consider the combination of one single source domain and one single target domain, and consider the generalization performance on the target domain accuracy after adaptation. In such situation, we remove the replay memory, the distillation loss term (Eq. 10) and the second term in the nominator of $\mathcal{L}_{\text{PCA}}$ (Eq. 9), as they force the model to achieve forgetting alleviation for previous domains and may impede the adaptation to the target domain. We conduct our stationary domain adaptation in PACS, and the results after adaptation are shown in Tables 9 Compared with other baselines, `RaTP` performs the best in most cases and achieves 4.4% higher average accuracy than the second best.

Table 6: Performance comparisons between ours and other baseline methods on Digits in average TDG, TDA, and FA under 10 different domain orders. We blue and bold **the best** performance, and bold **the second best**.

| Metric & Orders | | SHOT+ | SHOT++ | Tent | AdaCon | EATA | L2D | PDEN | Ours |
|---|---|---|---|---|---|---|---|---|---|
| TDG | Order 1 | 66.2 | 68.3 | 71.1 | **72.6** | 71.3 | 72.3 | 69.4 | **77.0** |
| | Order 2 | 78.0 | 78.2 | 72.9 | 75.8 | 71.5 | 78.1 | **78.4** | **79.5** |
| | Order 3 | 68.3 | 65.8 | 70.7 | 67.0 | 69.6 | **71.7** | 70.5 | **77.0** |
| | Order 4 | 49.1 | 52.0 | 52.2 | 53.2 | 53.7 | **62.3** | 60.4 | **72.0** |
| | Order 5 | 54.0 | 54.1 | 53.1 | 51.1 | 53.6 | **62.7** | 61.4 | **72.9** |
| | Order 6 | 72.3 | 75.2 | 76.9 | 75.8 | 77.8 | **78.2** | 76.8 | **81.0** |
| | Order 7 | 74.8 | 76.0 | 76.9 | 73.0 | 76.1 | **78.1** | 76.8 | **81.9** |
| | Order 8 | 73.9 | 72.6 | **79.3** | 76.9 | 77.9 | 78.0 | 77.3 | **81.3** |
| | Order 9 | 35.1 | 39.0 | 41.3 | 41.3 | 44.1 | **61.7** | **61.7** | **73.2** |
| | Order 10 | 38.6 | 41.7 | 45.9 | 46.3 | 44.2 | **65.5** | 63.8 | **71.7** |
| | Avg. | 61.0 | 62.3 | 64.0 | 63.3 | 64.0 | **70.9** | 69.7 | **76.8** |
| TDA | Order 1 | 84.0 | **87.5** | 71.5 | 77.4 | 76.8 | 85.9 | 81.9 | **89.7** |
| | Order 2 | **91.6** | **94.8** | 77.5 | 76.0 | 76.9 | 91.3 | 89.5 | 90.7 |
| | Order 3 | 81.2 | 79.9 | 70.7 | 75.8 | 76.4 | 85.9 | **86.2** | **87.8** |
| | Order 4 | 73.8 | **79.6** | 59.9 | 64.9 | 65.0 | 77.6 | 75.3 | **86.8** |
| | Order 5 | 79.7 | **84.9** | 59.5 | 65.3 | 65.8 | 79.3 | 78.3 | **87.5** |
| | Order 6 | 87.0 | **92.1** | 80.2 | 80.5 | 81.1 | 89.7 | 89.0 | **90.0** |
| | Order 7 | 89.9 | **91.2** | 80.9 | 82.1 | 83.2 | 87.6 | 85.2 | **91.6** |
| | Order 8 | 89.0 | **91.5** | 80.5 | 80.2 | 82.2 | 88.6 | 85.9 | **89.7** |
| | Order 9 | 48.4 | 48.8 | 48.7 | 55.7 | 55.4 | **74.2** | 70.9 | **85.9** |
| | Order 10 | 61.2 | 62.9 | 57.3 | 58.3 | 57.1 | **82.9** | 80.3 | **87.1** |
| | Avg. | 78.6 | 81.3 | 68.7 | 71.6 | 72.0 | **84.3** | 82.3 | **88.7** |
| FA | Order 1 | 60.0 | 67.1 | 67.8 | 75.2 | **76.2** | 75.2 | 71.4 | **83.8** |
| | Order 2 | 73.9 | 75.5 | 82.2 | 82.7 | **83.6** | 81.1 | 79.6 | **87.4** |
| | Order 3 | 70.7 | 71.2 | 72.9 | 80.4 | **85.5** | 85.1 | 81.9 | **90.1** |
| | Order 4 | 56.5 | 65.3 | 50.8 | 59.0 | 58.8 | **72.3** | 70.0 | **82.3** |
| | Order 5 | 77.0 | **79.1** | 61.4 | 71.7 | 71.2 | 74.9 | 73.9 | **85.2** |
| | Order 6 | 59.3 | 67.4 | **81.2** | 80.4 | 79.7 | 76.8 | 74.1 | **84.9** |
| | Order 7 | 62.2 | 71.0 | 79.8 | **82.1** | 80.9 | 77.6 | 76.1 | **84.7** |
| | Order 8 | 57.2 | 66.0 | 80.0 | **81.9** | 79.4 | 75.0 | 72.6 | **83.3** |
| | Order 9 | 25.1 | 30.0 | 33.1 | 56.8 | 61.8 | **72.5** | 68.5 | **85.1** |
| | Order 10 | 39.7 | 52.5 | 51.5 | 52.0 | 52.4 | **74.1** | 72.0 | **82.8** |
| | Avg. | 58.2 | 64.5 | 66.1 | 72.2 | 73.0 | **76.5** | 74.0 | **85.0** |

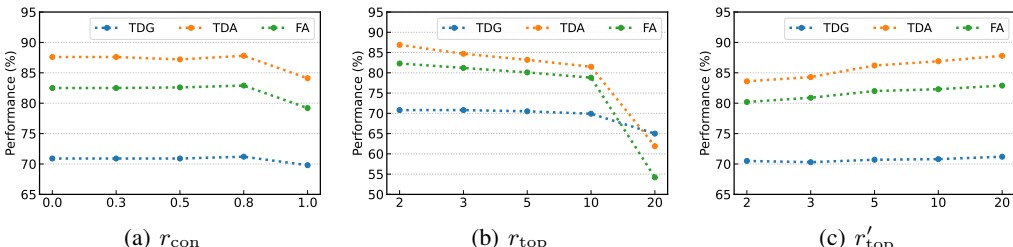

(a) $r_{\text{con}}$        (b) $r_{\text{top}}$        (c) $r'_{\text{top}}$

Figure 5: Sensitivity analysis of confidence threshold $r_{\text{con}}$ in `RandMix`; $r_{\text{top}}$ and $r'_{\text{top}}$ in `T2PL`.

Table 7: Performance comparisons between ours and other baseline methods on PACS in average TDG, TDA, and FA under 10 different domain orders. We blue and bold **the best** performance, and bold **the second best**.

| Metric & Orders | | SHOT+ | SHOT++ | Tent | AdaCon | EATA | L2D | PDEN | Ours |
|---|---|---|---|---|---|---|---|---|---|
| TDG | Order 1 | 69.4 | 70.4 | **75.5** | 75.2 | 75.1 | 74.0 | 73.7 | **76.6** |
| | Order 2 | 67.0 | 68.7 | 73.1 | 74.6 | 72.5 | **76.0** | 71.6 | **75.8** |
| | Order 3 | 67.8 | 63.3 | 75.6 | 75.9 | **76.1** | 72.8 | 73.5 | **76.6** |
| | Order 4 | 69.5 | 66.1 | **78.5** | 77.1 | 77.4 | 78.1 | 77.2 | **79.8** |
| | Order 5 | 61.1 | 62.2 | **81.6** | 74.6 | **78.3** | 74.6 | 73.5 | 78.1 |
| | Order 6 | 48.5 | 50.0 | 56.2 | 57.2 | **57.4** | 56.5 | 55.8 | **62.6** |
| | Order 7 | 36.6 | 43.2 | 52.5 | **55.4** | 54.3 | 54.9 | 52.0 | **56.8** |
| | Order 8 | 37.2 | 39.0 | 50.6 | 52.1 | 51.8 | **52.8** | 51.5 | **54.8** |
| | Order 9 | 53.1 | 52.7 | 54.3 | 57.3 | 48.2 | **62.0** | 60.9 | **73.8** |
| | Order 10 | 39.1 | 44.6 | **60.2** | 52.3 | 50.0 | 56.7 | 54.6 | **70.7** |
| | Avg. | 54.9 | 56.0 | 65.8 | 65.2 | 64.1 | **65.8** | 64.4 | **70.5** |
| TDA | Order 1 | 86.7 | **89.4** | 84.0 | 85.8 | **86.7** | 84.5 | 83.7 | 83.5 |
| | Order 2 | **87.8** | **89.4** | 82.0 | 81.6 | 85.3 | 83.6 | 83.3 | 84.5 |
| | Order 3 | **88.7** | **88.8** | 82.5 | 82.8 | 85.6 | 82.9 | 79.9 | 84.2 |
| | Order 4 | **89.2** | **91.2** | 88.2 | 88.7 | 89.2 | 84.6 | 83.0 | 86.8 |
| | Order 5 | 85.2 | 85.4 | **88.6** | 86.4 | **88.2** | 80.1 | 78.2 | 85.5 |
| | Order 6 | **83.1** | **85.3** | 75.7 | 78.6 | 79.2 | 75.5 | 74.0 | 82.3 |
| | Order 7 | 66.9 | 69.9 | **74.4** | 74.0 | 73.0 | 71.3 | 71.6 | **76.6** |
| | Order 8 | 64.0 | 68.8 | 72.5 | **73.9** | 72.3 | 68.5 | 69.8 | **74.9** |
| | Order 9 | **91.5** | **92.2** | 69.6 | 77.8 | 73.0 | 83.0 | 82.5 | 89.7 |
| | Order 10 | 75.9 | **83.2** | 69.8 | 69.7 | 70.4 | 74.4 | 72.1 | **90.1** |
| | Avg. | 81.9 | **84.4** | 78.7 | 79.9 | 80.3 | 78.8 | 77.8 | **83.8** |
| FA | Order 1 | 73.0 | 78.6 | 89.5 | **90.7** | **91.4** | 85.6 | 85.2 | 87.8 |
| | Order 2 | 72.4 | **82.3** | 79.5 | 77.7 | **83.7** | 80.6 | 77.4 | 77.9 |
| | Order 3 | 81.8 | 78.9 | 88.5 | **89.5** | **90.5** | 84.8 | 78.7 | 87.5 |
| | Order 4 | 76.9 | 77.6 | 83.3 | **84.5** | **87.7** | 77.5 | 77.1 | 82.5 |
| | Order 5 | 82.9 | 86.1 | **89.0** | 88.0 | **90.8** | 76.7 | 76.5 | 84.2 |
| | Order 6 | 79.6 | **84.3** | 81.4 | 80.7 | 83.5 | 71.0 | 70.9 | **86.2** |
| | Order 7 | 65.3 | **80.5** | 78.0 | 78.0 | 74.4 | 75.7 | 75.4 | **79.7** |
| | Order 8 | 58.3 | **83.5** | 73.3 | **74.0** | 73.3 | 72.6 | 72.1 | 73.4 |
| | Order 9 | 86.5 | **88.8** | 74.1 | 79.0 | 76.6 | 78.0 | 78.3 | **87.5** |
| | Order 10 | 72.0 | **89.5** | 73.1 | 73.7 | 74.3 | 73.4 | 71.4 | **89.6** |
| | Avg. | 74.9 | **83.0** | 81.0 | 81.6 | 82.6 | 77.6 | 76.3 | **83.6** |

Table 8: Performance comparisons between ours and other baseline methods on DomainNet in average TDG, TDA, and FA under 10 different domain orders. We blue and bold **the best** performance, and bold **the second best**.

| Metric & Orders | | SHOT+ | SHOT++ | Tent | AdaCon | EATA | L2D | PDEN | Ours |
|---|---|---|---|---|---|---|---|---|---|
| TDG | Order 1 | 46.9 | 45.5 | **52.1** | 51.3 | 51.2 | 49.6 | 48.0 | **51.9** |
| | Order 2 | 52.2 | 50.0 | 31.0 | 52.7 | 55.2 | **55.6** | 51.2 | **57.4** |
| | Order 3 | 53.6 | 53.3 | 58.4 | 53.7 | **58.7** | 52.8 | 51.1 | **59.4** |
| | Order 4 | 40.8 | 41.9 | 50.6 | **51.3** | 51.1 | 48.2 | 47.0 | **54.7** |
| | Order 5 | 48.4 | 49.6 | 52.8 | 53.0 | 52.8 | **53.1** | 51.0 | **56.0** |
| | Order 6 | 34.0 | 35.3 | 33.1 | 32.9 | 33.6 | 36.5 | **37.2** | **46.7** |
| | Order 7 | 23.2 | 25.7 | **35.4** | 32.9 | 34.4 | 32.2 | 30.2 | **41.5** |
| | Order 8 | 59.2 | 59.9 | 61.0 | **62.0** | **62.0** | **62.1** | 61.4 | 61.8 |
| | Order 9 | 58.7 | 59.3 | 61.3 | 61.6 | **63.3** | 59.4 | 59.9 | **63.7** |
| | Order 10 | 56.2 | 60.1 | 41.2 | **61.3** | 59.0 | 57.4 | 56.4 | **62.3** |
| | Avg. | 47.3 | 48.1 | 47.7 | 51.3 | **52.1** | 50.7 | 49.3 | **55.5** |
| TDA | Order 1 | **68.4** | **70.5** | 59.0 | 60.4 | 60.0 | 59.9 | 60.8 | 68.1 |
| | Order 2 | **69.7** | 66.2 | 28.9 | 66.2 | 65.8 | 56.5 | 54.2 | **69.6** |
| | Order 3 | **72.6** | **73.4** | 65.6 | 68.6 | 69.4 | 54.8 | 52.7 | 70.3 |
| | Order 4 | 51.3 | 53.5 | 54.6 | 52.2 | 52.9 | **57.3** | 55.2 | **60.3** |
| | Order 5 | **68.5** | **70.9** | 60.3 | 60.3 | 58.7 | 56.9 | 55.7 | 68.2 |
| | Order 6 | 63.1 | **65.3** | 51.8 | 52.4 | 55.0 | 49.3 | 50.2 | **65.1** |
| | Order 7 | 47.7 | 48.1 | 50.0 | 50.8 | **51.7** | 41.7 | 40.4 | **56.8** |
| | Order 8 | 72.8 | **73.2** | 67.6 | 71.6 | 70.7 | 64.0 | 63.4 | **75.5** |
| | Order 9 | **74.2** | **75.9** | 67.6 | 71.3 | 69.3 | 61.9 | 62.2 | 74.1 |
| | Order 10 | **71.9** | **72.1** | 31.0 | 67.9 | 71.2 | 59.9 | 61.0 | 67.8 |
| | Avg. | 66.0 | **66.9** | 53.6 | 62.2 | 62.5 | 56.2 | 55.6 | **67.6** |
| FA | Order 1 | 61.4 | 66.5 | **67.4** | 67.0 | 64.3 | 63.7 | 61.1 | **67.6** |
| | Order 2 | 64.5 | **68.9** | 34.1 | 62.6 | 65.8 | 48.9 | 46.3 | **67.7** |
| | Order 3 | 62.9 | 67.7 | 65.6 | 66.3 | **69.2** | 45.2 | 43.1 | **68.1** |
| | Order 4 | 42.1 | **65.4** | 56.4 | 53.3 | 52.7 | 41.5 | 39.5 | **56.5** |
| | Order 5 | 60.9 | **68.5** | 58.0 | 56.6 | 57.4 | 51.2 | 48.6 | **65.4** |
| | Order 6 | 61.1 | **66.3** | 52.4 | 49.4 | 54.8 | 48.0 | 46.0 | **62.8** |
| | Order 7 | 42.8 | **51.7** | 48.5 | 48.5 | 47.7 | 37.2 | 36.0 | **53.7** |
| | Order 8 | 61.6 | 67.5 | 71.6 | **72.8** | **73.5** | 58.8 | 55.1 | 72.0 |
| | Order 9 | 67.4 | **77.3** | 76.8 | 76.1 | 76.0 | 66.5 | 65.6 | **79.6** |
| | Order 10 | 60.4 | **69.6** | 30.4 | 65.5 | **66.3** | 61.4 | 60.9 | 62.9 |
| | Avg. | 58.5 | **66.9** | 56.1 | 61.8 | 62.8 | 52.2 | 50.2 | **65.6** |

Table 9: Comparison of domain adaptation for stationary source-target pairs on PACS. We blue and bold **the best** average performance, and bold **the second best**.

| Method | P→A | P→C | P→S | A→P | A→C | A→S | C→P | C→A | C→S | S→P | S→A | S→C | Avg. |
|---|---|---|---|---|---|---|---|---|---|---|---|---|---|
| SHOT | 87.2 | 85.4 | 33.8 | 99.3 | 87.8 | 44.4 | 99.0 | 91.9 | 64.9 | 53.5 | 84.9 | 89.8 | 76.8 |
| SHOT+ | 90.4 | 87.9 | 38.8 | 99.5 | 88.8 | 41.9 | 98.9 | 92.6 | 67.2 | 52.2 | 84.3 | 89.5 | 77.7 |
| SHOT++ | 93.1 | 89.1 | 38.8 | 99.6 | 90.1 | 41.3 | 99.1 | 93.6 | 68.2 | 53.1 | 87.2 | 91.7 | 78.4 |
| Tent | 77.0 | 64.9 | 55.5 | 97.4 | 76.7 | 63.7 | 95.4 | 82.8 | 73.5 | 55.3 | 59.1 | 68.0 | 72.4 |
| AdaCon | 88.2 | 85.3 | 56.6 | 99.5 | 86.8 | 67.7 | 97.7 | 91.5 | 75.0 | 61.3 | 82.7 | 75.3 | **80.6** |
| EATA | 79.2 | 72.7 | 59.0 | 97.2 | 80.0 | 72.7 | 95.7 | 84.8 | 79.7 | 55.3 | 60.0 | 68.7 | 75.4 |
| Ours | 92.5 | 83.2 | 62.4 | 99.5 | 88.0 | 59.0 | 98.9 | 92.6 | 64.8 | 99.2 | 92.8 | 86.7 | **85.0** |

