# OpenReview forum: "Deja Vu: Continual Model Generalization for Unseen Domains"
_ICLR.cc/2023/Conference — ICLR 2023 poster_

### Official Review · Reviewer_CHgo · 2022-10-23

**Confidence:** 5
**Clarity, Quality, Novelty And Reproducibility:** 1.	Writing is clear and easy to follo…
**Correctness:** 4
**Technical Novelty And Significance:** 4
**Empirical Novelty And Significance:** 4
**Recommendation:** 8

**Strength And Weaknesses:**

Strength:

1.	The idea of considering performance before adaptation in domain continual learning setting is novel and practical in the real world.

2.	The training-free RandMix method can keep the cost low and achieve high generalization ability, which is very important in this continual domain generalization setting. In many cases, scenarios that require the use of this setting, such as surveillance cameras, have limited computing power.

3.	Only using a subset of features and introducing kNN for final selection in T2PL can filter out features that can cause side effects. I think the effect of this method will be more obvious when the data noisy is relatively high.

4.	The proposed Prototype Contrastive Alignment is novel. Instead of modeling the distribution of previous tasks and then aligning the features in existing works, PCA provides a new and more efficient way to align previous and current features by saving a small part of network parameters and reducing the domain adaptivity gap. Furthermore, this loss function is easy to be adapted to other frameworks.

5.	The experiments show high effectiveness compared to other SOTA methods, especially when target domain is much more complicated than source domain, e.g., SVHN in Digital dataset.

Weaknesses:

1. The idea of RandMix looks like a simple combination of DSU and L2D. Not sure my understanding is correct or not.

2. The paper introduces the kNN in T2PL without detailed explanations of why it can work. I think more analysis about it needs to be provided.

3. It seems that the domain order is an important factor that influences the overall performance. In main paper, most of the average performance of three metrics is 5-10% better than the second best. But in appendix, it’s only 1-2% better or even worse than baseline. Can the author(s) explain the reason why?

4. In table 1, the drop in accuracy on SVHN is too large for all source free adaptation methods and makes me doubt the accuracy of the experiments. Can the author(s) explain the reason and I’m also checking the experiments results in related works.


**Summary Of The Paper:**

This work is very interesting. It tries to simultaneously address domain generalization, domain adaptation and catastrophic forgetting problem when the learning model needs to tackle continual domain shifts over time, which is called Continual Domain Shift Learning (CDSL). To solve this, this work proposes a framework – Twofer, which consists of three major modules. Specifically, there is a data augmentation module that can generate more additional data to help enhance domain generalization ability. Then a novel KNN-based pseudo labeling technique is proposed for giving more accurate supervision that can be used in later adaptation. Finally, a modified prototype contrastive loss is used to align features among different domains. The proposed twofer is tested on various dataset and domain orders, with the comparison with extensive state-of-the-art baselines from continual domain adaptation, test-time adaptation, source-free domain adaptation, single domain generalization, multiple domain generalization, and even unified domain adaptation and generalization. According to experiment results, twofer can outperform these baselines on three metrics that are related to domain generalization, adaptation, and forgetting.

**Summary Of The Review:**

In summary, the setting is practical and the proposed methods are novel and effective. I thus recommend this paper to appear at ICLR.

---

> ### Author Response · Authors · 2022-11-18
> **Response to Reviewer CHgo [1/2]**
>
> >1. The idea of RandMix looks like a simple combination of DSU and L2D. Not sure my understanding is correct or not.
>
> While our RandMix module is inspired by L2D, the way we introduce diversity into the data augmentation is different. L2D mainly relies on managing the Mutual Information (MI) in hidden space to obtain diversified distributions. There is no restriction in L2D for expanding the data augmentation, which may hurt the model generalization ability because the augmented data has too little shared semantics with the original data when the expansion is large. In contrast, RandMix is training-free and we obtain diversity by introducing variance-restricted random noise. Moreover, the model parameters in L2D are fixed for each mini-batch, while RandMix initializes model parameters every time when there is a new mini-batch, which can introduce more diversities. After uncertainty modeling on representations extracted by the feature extractor, DSU injects uncertainty shifts to achieve representation augmentation, while our approach uses noise to produce sample augmentation. The representation augmentation achieved by DSU is built on the backbone feature extractor that can be used to model the feature uncertainty in terms of domain generalization. However, if we apply DSU on the representation extracted by a single convolutional layer, i.e., the encoder of RandMix, the representation is too low-level to be used for uncertainty modeling. As a result, it is difficult to combine DSU and L2D for achieving effective data augmentation like RandMix.
>
> >2. The paper introduces the kNN in T2PL without detailed explanations of why it can work. I think more analysis about it needs to be provided.
>
> Thanks for the comment. In cross-domain learning, samples from different domains are highly overlapped, and giving pseudo labels by comparing samples with centroids of different classes only once is risky. In our kNN-based pseudo labeling, kNN can decentralize the risk of misclassification in assigning pseudo labels by a single comparison between class centroids and data samples. Besides, our T2PL works as a voting system that can encourage the features of certain samples to have the same labels as their neighbors and can be interpreted as a kind of local structure clustering as mentioned in G-SFDA (Yang et al. ICCV’2021). As a result, T2PL is more suitable for clustering samples that are highly overlapped than center-based approaches. In the revision, we have added more explanations about kNN in T2PL based on the above reasoning.
>
> >3. It seems that the domain order is an important factor that influences the overall performance. In main paper, most of the average performance of three metrics is 5-10% better than the second best. But in appendix, it’s only 1-2% better or even worse than baseline. Can the author(s) explain the reason why?
>
> Thanks for the question. Domain order is indeed a very important factor in continual learning settings. In stationary domain adaptation settings, the results for different combinations of source and target domains can vary a lot. In the continual domain adaptation settings, the difference could be even larger as the difference accumulates along with the continual domain shift. Such difference is shown in various continual domain adaptation papers like LDAuCID (Rostami, 2021) and G-SFDA (Yang et al. ICCV’2021).
>
> In the revision, we have carried out additional experiments to assess the impact of domain order on our approach and baselines, and report the results in Section C of the Appendix. Specifically, we randomly shuffle the domains for all three datasets and try to balance the possibility of taking each domain as the starting one to increase the representativeness (Table 5 in the Appendix). We conduct experiments for 10 random orders for each dataset (the most we can do within the rebuttal time frame), and report the average TDG, TDA, and FA under these 10 domain orders. The experimental results demonstrate similar observations as in the original submission, i.e., our approach significantly outperforms state-of-the-art baseline methods in TDG, and achieves comparable performance in TDA and FA. Please refer to Tables 9, 10, and 11 in the Appendix for detailed results.

---

> > ### Comment · Reviewer_CHgo · 2022-11-20
> > **Response to Authors**
> >
> > I'm satisfied with the response. So I raise my original score.

---

> > > ### Author Response · Authors · 2022-11-21
> > > **Thanks to Reviewer CHgo**
> > >
> > > Thank you very much for checking our response and thanks again for the initial comments. They are very helpful for our improvement of the paper.

---

> ### Author Response · Authors · 2022-11-18
> **Response to Reviewer CHgo [2/2]**
>
> >4. In table 1, the drop in accuracy on SVHN is too large for all source free adaptation methods and makes me doubt the accuracy of the experiments. Can the author(s) explain the reason and I’m also checking the experiments results in related works.
>
> In the Digits dataset for Table 1, SVHN is actually the most complex domain. For example, the results in SHOT (Liang et al. ICML 2020) show that the generalization accuracy on SVHN before the adaptation is quite low. Thus, its pseudo-labeling performs badly and further results in poor adaptation results. The difficulty of adapting to SVHN is also confirmed in Tent (Wang et al. ICLR 2021): In the authors’ rebuttal response (https://openreview.net/forum?id=uXl3bZLkr3c), they confirm that after adapting to SVHN from MNIST, the accuracy dropped by about 9%. Fortunately, this problem can be solved by applying RandMix to SHOT as shown in the ablation study.
>
> >5. Some average accuracy in Table 6 is miscalculated, please check them again.
>
> Thanks for your careful checking. We have double-checked the table and corrected the typos.

---

### Official Review · Reviewer_sJyy · 2022-10-24

**Confidence:** 3
**Correctness:** 2
**Technical Novelty And Significance:** 2
**Empirical Novelty And Significance:** Not applicable
**Recommendation:** 5

**Clarity, Quality, Novelty And Reproducibility:**

Clarity, Quality: Poor
Novelty: Not sure.
Reproducibility: Good. The code is provided.

**Strength And Weaknesses:**

Strength:
1. The motivation is new and seems to be practical proposed by the authors;
2. The experimental results are superior.

Weaknesses:
1. The writing of the manuscript is hard to follow, some definitions are unclear, i.e., what is distinguishability of data samples in 4th line of P5? Moreover, some formulas have better expression but not the adopted ones, which takes me a lot of time to grab the meaning. The choice of notation is unprofessional.
2. The motivation of random mixup augmentation is weak. As far as I am concerned, deploying variant noise in training does harm to stability of training. Then, how it improves the generalization is hard to explain. Moreover, there are an ocean of noise generation method, why Eq. 2 is better?
3. There lacks necessary explaination to Sec2.3 and the caption and content of Fig2 are unclear.



**Summary Of The Paper:**

The manuscript proposes to tackle the continual domain shift learning issue, in which the model is trained on a source domain and serveral unlabeled target domains. The authors highlight the pain points of the issue are concluded with three: 1. model generalization on 'before and during' training doamins(TDG); 2. better usage of unlabeled target domains(TDA); 3. a commonplace talk of an old scholar which is Catastrophic Forgetting. To address the issues, a Twofer method is proposed, which is composed by three components: 1. Random Mixup Augmentation; 2. Top^2 pseudo labeling; 3. prototype contrative alignment. Finally, extensive experiments are conducted.



**Summary Of The Review:**

In summary, the biggest issue to me is the writing. I cannot get the motivation of the proposed modules. To this end, I think the manuscript needs further polish.

---

> ### Author Response · Authors · 2022-11-18
> **Response to Reviewer sJyy**
>
> >1. The writing of the manuscript is hard to follow, some definitions are unclear, i.e., what is distinguishability of data samples in 4th line of P5? Moreover, some formulas have better expression but not the adopted ones, which takes me a lot of time to grab the meaning. The choice of notation is unprofessional.
>
> Thanks very much for the comment. We have rewritten some of the descriptions and definitions to make them more clear. In particular, the distinguishability of data samples describes the possibility of correct pseudo-labeling, and we have revised the writing in the paper (pages 4 and 5). For the formulas and notations, we have carefully compared ours with other related works and made corresponding modifications to improve their readability.
>
> >2. The motivation of random mixup augmentation is weak. As far as I am concerned, deploying variant noise in training does harm to stability of training. Then, how it improves the generalization is hard to explain. Moreover, there are an ocean of noise generation method, why Eq. 2 is better?
>
> In domain generalization (DG) problems, target domain data is unavailable during training, and only source domain data is provided. In such a setting, DG requires the trained model to generalize well in target domains, which is often a tough out-of-distribution generalization task. Therefore, many DG studies apply data augmentation to expand the distribution of given source domain data and help improve model generalization. Such data augmentation approaches aim to generate data with different domain styles, e.g., changing the color and shape of a picture. Then training on the augmented data can help the model learn to capture the shared semantics among different domains, e.g., whether certain cartoon cats and real cats belong to the same class of cat. In this way, data augmentation can improve the model generalization ability.
>
> We do agree that variant noise may make the training unstable and hurt the model performance, and in fact, we observed that blindly deploying augmentation techniques to the original data could be detrimental to the model's generalization ability. Thus, we design RandMix as in Eq. (2) to restrict the introduced noise within a reasonable range, i.e., not too large or too small. First, L2D has proven that augmentation data generated by convolution and transposed-convolution autoencoders has bounded domain discrepancy to the original data, but the discrepancy is small. L2D thus adopts mutual information maximization to increase the domain discrepancy, however, such unrestricted discrepancy increase reduces the shared semantics between the original and the augmentation data. Other data augmentation approaches also have similar issues. In RandMix, we apply AdaIN to inject noise for increasing the domain discrepancy, but the increase is bounded because the AdaIN parameters are re-initialized from a normalized distribution for each mini-batch. As the input noise of AdaIN is also from a normalized distribution, the actually introduced noise has a smaller variance. In other words, we use AdaIN to expand the domain discrepancy between the original and the augmentation data to a reasonable and bounded range. Besides, we notice that the diversity of augmentation data generated by existing approaches is insufficient. Thus, we use random weighted mixup to combine augmentations from a series of autoencoders as in Eq. (2). In summary, RandMix addresses issues of both unrestricted discrepancy enlargements and diversity insufficiency, which are major shortcomings of current state-of-the-art DG data augmentation methods. In addition, RandMix is training-free, thus a better choice for solving DG problems.
>
> >3. There lacks necessary explaination to Sec2.3 and the caption and content of Fig2 are unclear.
>
> Thanks for the comment. In the revision, we have added more explanations in Section 2.3, mainly on the reason why existing pseudo-labeling approaches are not good enough in cross-domain scenarios, and how our T2PL can provide a better solution. We have also re-drawn Figure 2 and revised its caption to improve its readability and better illustrate the issue of adaptivity gap enlargement that is caused by applying regular prototype learning to align different domains.

---

> ### Author Response · Authors · 2022-11-21
> **A Gentle Reminder of Further Feedback**
>
> Thank you again for the first round of review feedback. We hope our response is able to address your comments related to the writing readability and further explanation of our data augmentation. We take this as a great opportunity to improve our work and shall be grateful for any additional feedback you could give us.

---

> ### Author Response · Authors · 2022-11-29
> **With the hope that our response addresses your concerns**
>
> Dear Reviewer sJyy,
>
> The conclusion of discussion period is closing, and we eagerly await your response. We greatly appreciate your time and effort in reviewing this paper and helping us improve it.
>
> We have provided detailed responses to every one of your concerns. Please help us to review our responses once again and kindly let us know whether they fully or partially address your concerns and if our explanations are in the right direction. We shall be grateful for any additional feedback you could give us.
>
> Kind Regards,
>
> Authors of Paper477

---

> ### Author Response · Authors · 2022-12-07
> **Would Appreciate Further Feedback on the Rebuttal with Reviewer sJyy**
>
> Dear Reviewer sJyy,
>
> Thank you again for the initial comments. As the discussion period is closing soon, we would greatly appreciate any feedback on our rebuttal. We fully understand that you may be busy at this time, but hope that you could kindly have a quick look at our responses and assess whether they have addressed your comments and warrant an update to the rating. We would also welcome any additional feedback and questions.
>
> Best Regards,
>
> Authors of Paper477

---

### Official Review · Reviewer_jYsN · 2022-10-29

**Confidence:** 4
**Correctness:** 3
**Technical Novelty And Significance:** 2
**Empirical Novelty And Significance:** 3
**Recommendation:** 5

**Clarity, Quality, Novelty And Reproducibility:**

*Clarity*:

- The introduction's discussion of the "unfamiliar period" does reframe the claims in the abstract for the better, by focusing the paper on accuracy pre-adaptation and post-adaptation.
  However, this framing should come first, and be reflected in the abstract and first part of the introduction, before claiming that the proposed TwoFer "significantly outperforms" everything and is "envisioned as a significant milestone".
  These are not precise nor productive claims as written.
  Furthermore, in talking about the unfamiliar period and the metrics TDG, TDA, and FA, this work could align with the literature on continual learning and discuss "forward transfer" and "backward transfer" instead of introducing custom terminology.
- The main results (Tables 1, 2, 3) obscure the compared methods with custom abbreviations (TEN instead of Tent, CoT instead of CoTTA, etc.).
  While these can be decoded and checked against the text, it would be faster to read if the original names and abbreviations were kept.

*Quality*:

- The main results in Tables 1 & 2 appear to show large gains in average accuracy across domains for the three metrics of accuracy before/during/after a given domain.
- However, as already noted this evaluation is not entirely standard, and the standard evaluation is not included in the results to first establish performance on a known setting.

*Novelty*:

- Contrary to its claim, Twofer is not the first work to address continual adaptation or "continual domain shift learning".
  [Efficient Test-Time Model Adaptation without Forgetting](https://arxiv.org/abs/2204.02610) at ICML'22 and [Continual Test-time Domain Adaptation](https://arxiv.org/abs/2203.13591) CVPR'22 both address adaptation to a sequence of domains, did so well before the submission deadline for ICLR'23, and report results on larger-scale and more difficult datasets.
  The earliest instance of learning on non-stationary domains is most likely [Continuous Manifold Based Adaptation for Evolving Visual Domains](https://openaccess.thecvf.com/content_cvpr_2014/html/Hoffman_Continuous_Manifold_Based_2014_CVPR_paper.html), although it works in the unsupervised domain adaptation setting, but that does not make it unrelated.
- The proposed pseudo-labeling approach is in essence clustering while filtering the predictions by certainty, but such filtering or thresholding is common for pseudo-labeling. See the PAMI edition of SHOT (Liang et al. 2021) called SHOT++ or EATA (cited above in this review) for examples. In particular SHOT++ has an adaptive threshold based on the distribution of confidences, which is related to filtering a fixed percentage as done by T2PL.
- The use of parameters as prototypes is not novel to this work, and has been done not only by the cited Saito et al. 2019, but goes back to at least Imprinted Weights by Qi et al. CVPR'18.

*Reproducibility*:

- While there are many parts to the method, there is code provided in the supplementary materials. (That is, it is promised in the main text, but this was not verified as part of the review.)

**Strength And Weaknesses:**

*Strengths*:

- The setting emphasizes adaptation across multiple domains, in sequence, and the need for more metrics to do so. This setting measures not only accuracy _during adaptation_ to a given domain, but _before_ to measure domain generalization as well as _after_ to measure resilience to forgetting.
- The baselines are drawn broadly from source-free adaptation and domain generalization (but they are not comprehensive, see weaknesses), and for some baselines such L2D, PDE, and PCL there is an attempt to equip them for the proposed setting by providing the same exemplar memory as the proposed method.
- The augmentation model, building on domain diversification (Wang et al. 2021) and AdaIN (Karras et al. 2019), proves effective for the proposed method as well as baslines, and significantly impoves accuracy for adaptation across digits datasets (Figure 4a).
- The ablation study (Section 3.2 and Figure 4) justifies augmentation by randmix, the pseudo-labeling by T2PL, and most of all the PCA loss. However, the ablation is only on Digits, which is the simplest dataset, and it would be more rigorous to repeat these experiments for PACS and DomainNet too in order to check for consistent effects.

*Weaknesses*:

- The proposed setting depends on the order of domains, but only two orders are shown in the main paper, and just one additional order in the appendix. The order in the Appendix for PACS and DomainNet shows _no or little improvement_ for Twofer, while Digits does still improve, which suggests the possibility that order can be quite important. This small sampling of orders is unlikely to give a non-noisy estimate of performance.
- The framing is neither accurate nor cordial and collegial. This paper does not begin the study of continual shift, counter to the claims of the abstract and introduction. Rather it focuses its metrics on particular phases of adaptation, and emphasizes (1) rapid learning at the beginning of a domain and (2) less forgetting on past domains.
- The evaluation protocol differs from the compared work, in defining its own splits of the data (see "Experiment Settings"), but how exactly the splits are defined is not clear. Alternative evaluations can be necessary to make a point, but such oddball evaluations should be paired with the standard evaluations for fair comparison and sanity checking of the results.
- There is missing related work on test-time adaptation with more efficient updating and slower forgetting: Efficient Test-Time Model Adaptation Without Forgetting (EATA) at ICML'22. As this prior work addresses key claims of TwoFer, it needs to be compared with in the experimental evaluation. It should likewise be discussed should its contributions and technical details intersect with the claims in the proposed method.
- There is missing related work on source-free adaptation: AdaContrast (Chen et al. CVPR'22) and SHOT++ (Liang et al. PAMI'21). This requires discussion and comparison, as results are reported for DomainNet and other standard benchmarks.
- The proposed method heavily relies on data augmentation, which makes it more domain specific than other methods, which perform none or little, like Tent or EATA for example.



**Summary Of The Paper:**

This work addresses adaptation to a sequence of discrete domains where accuracy before, during, and after adaptation is desired.
Accuracy before processing a domain measures domain generalization, accuracy during or immediately following a domain measures adaptation, and accuracy after a domain measures forgetting (or remembering).
This setting is named continual domain shift learning (CDSL) in this work, and presented as a new problem statement because of (1) the changing of the domains and (2) the metrics before/during/after the domain is adapted on.
The proposed method, TwoFer, incorporates elements of domain adaptation and domain generalization, and its most important components are agressive data augmentation (RandMix), a pseudo-labeling scheme that clusters and filters predictions to keep only a certain percentile, and a prototype representation of classes in the target domains (prototype contrastive alignment or PCA).
Each component is partly-tuned to its use here, but all of them have a strong relation to prior work that introduced it.
Experiments cover three benchmarks that are common to domain adaptation and generalization: digits, PACS, and DomainNet.
However, the evaluation protocol differs from the standard protocol, both in the desired measurement of sequences of domains, but also in a different sampling of splits.
Since the results are sensitive to the order of the domains adapted on, two orders are evaluated in the main paper for each dataset (with one more each in the appendix).
Baselines cover domain adaptation, in particular source-free and test-time adaptation, as well as domain generalization.
Ablations check the separate contribution of the data augmentation, pseudo-labeling, and prototypical loss components of the method, and analyze alternative choices or applications of each component.



**Summary Of The Review:**

The dismissal of prior work and the divergent evaluation w.r.t. the established protocol used by the compared methods raises the possibility of serious experimental error.
It may be that everything is sound, but the rebuttal needs to clarify exactly how the evaluation was done to be sure. Ideally, the rebuttal would report results in the standard experimental setup for a fair comparison to accompany the main results of the paper in Tables 1 & 2.
Without this information, it is hard to gauge the correctness and empirical significance of the submission.
I am open to counterevidence however, and will reconsider this submission in light of response and discussion.

*For Rebuttal*

Major

- Please report results in the standard evaluation protocols for PACS and DomainNet. In particular, please use the standard splits, and include results with stationary/non-continual/episodic domains. By comparing in the established setting, the experiments would establish that Twofer is at least as good there, and then the experiments could should improvement in the proposed protocol.
- Please analyze the sensitivity of TwoFer and the baselines to domain order. If the variance due to order is larger than the reported gains, then the experiments may have inadvertently chased noise.
- Please relate TwoFer to EATA and AdaContrast and report any comparisons, if any comparable comparisons are possible given the set of experiments in the submission.
- Please clarify the novelty or lack thereof in the setting, compared to the missing related work raised, and re-articulate the contributions of CDSL vs. other studies of continual shift in existing work like CoTTA and EATA.

Minor

- Was DomainNet filtered according to the evaluation protocol in prior work like Saito et al. 2019 and Chen et al. 2022 (AdaContrast)?

**Update after Response**

The substantial responses and experiments address concerns about experimental validity concerning data splits, domain ordering, and missing benchmarks. As such I am raising my score to 5 (borderline reject) to acknowledge these improvements while still urging the authors to consider the framing and organization of this work so that it is more comprehensible to an audience that spans adaptation, continual learning, and robustness. I have accordingly raised the marks for correctness and empirical novelty/significance as well.

---

> ### Author Response · Authors · 2022-11-18
> **Response to Reviewer jYsN [1/5]**
>
> **Weaknesses:**
>
> >1. The proposed setting depends on the order of domains, but only two orders are shown in the main paper, and just one additional order in the appendix. The order in the Appendix for PACS and DomainNet shows no or little improvement for Twofer, while Digits does still improve, which suggests the possibility that order can be quite important. This small sampling of orders is unlikely to give a non-noisy estimate of performance.
>
> Thanks very much for the comment. In the original submission, we followed some of the previous studies of continual domain adaptation [1, 2] to conduct experiments for two domain orders. And we also included an additional order in the Appendix. We do agree that the domain order could have a significant impact on the performance, and is an important topic in continual learning, e.g., in task-agnostic continual learning [3]. Our proposed Prototype Contrastive Alignment has in fact taken some impact of different domain orders into account, by addressing the cases in which the adaptivity gap changes significantly (i.e., when neighboring domains in the order have sharp domain discrepancy).
>
> In the revision, to further assess the impact of domain order, we have carried out a set of additional experiments and report the results in Section C.1 of the Appendix. Specifically, we randomly shuffle the domains for all three datasets and try to balance the possibility of taking each domain as the starting one to increase the representativeness (Table 5 in the Appendix). We conduct experiments for 10 random orders for each dataset (the most we can do within the rebuttal time frame), and report the average TDG, TDA, and FA under these 10 domain orders. The experimental results demonstrate similar observations as in the original submission, i.e., our approach significantly outperforms state-of-the-art baseline methods in TDG, and achieves comparable performance in TDA and FA. Please refer to Tables 9, 10, and 11 in the Appendix for detailed results.
>
> [1] Qin Wang, et al. Continual test-time domain adaptation. CVPR 2022.
>
> [2] Shixiang Tang, et al. Gradient regularized contrastive learning for continual domain adaptation. AAAI 2022.
>
> [3] Rajasegaran J, et al. itaml: An incremental task-agnostic meta-learning approach. CVPR 2020.
>
> >2. The framing is neither accurate nor cordial and collegial. This paper does not begin the study of continual shift, counter to the claims of the abstract and introduction. Rather it focuses its metrics on particular phases of adaptation, and emphasizes (1) rapid learning at the beginning of a domain and (2) less forgetting on past domains.
>
> Thanks for the comment. We have made changes throughout the paper, particularly in the Abstract and the Introduction, to clarify the focus and contributions of our approach. In particular, we emphasize that our main focus is to improve the model performance on new target domains before and during their training, in what we call the “Unfamiliar Period”; and the experimental results demonstrate that our approach can indeed significantly improve this objective, namely the target domain generalization (TDG) performance, over previous state-of-the-art methods. In addition, while not being the main focus, our approach also achieves comparable performance as those previous methods in model target domain adaptation (TDA) and forgetting alleviation (FA) capabilities. This balanced performance across TDG, TDA, and FA will hopefully help the adoption of our approach in practical scenarios.
>
> >3. The evaluation protocol differs from the compared work, in defining its own splits of the data (see "Experiment Settings"), but how exactly the splits are defined is not clear. Alternative evaluations can be necessary to make a point, but such oddball evaluations should be paired with the standard evaluations for fair comparison and sanity checking of the results.
>
> Thanks for the comment. In the original submission, we followed the recommended setting in [4] and randomly split 80% of the data for each domain of each dataset as the training set, while the rest 20% was the testing set. The reason for such a choice is that while compared works [5, 6] used the entire set of training data to test the adaptation performance in target domains, we think that such settings may not be practical enough in CDSL as the model may be adapted to highly overfit the training data and perform badly on unseen testing data, which is mentioned in [4].
>
> Nevertheless, we agree that it would be good to include results for the standard setting in the compared works [5, 6]. In the revision, we have added such experiments for all three datasets and presented the results in Section E of the Appendix (Tables 13, 14, 15). From the results, we can observe that our approach still works effectively and outperforms baselines in TDG, while achieving comparable performance in TDA and FA.

---

> > ### Comment · Reviewer_jYsN · 2022-11-30
> > **Thank you for the point-by-point response.**
> >
> > I first want to acknowledge the timely and thorough response which discusses each of the issues raised in this review as weaknesses and points for rebuttal. In summary, the submission has improved (by better framing its topic and providing results that are more comparable and less noisy), but issues remain. Most importantly, the experiments vary in their comparability across prior work and the experiments seem to selectively report results by only including certain conditions (like orders of domains) or comparisons.. Taking the response and discussions into account, I am currently maintaining my recommendation, as further steps are needed to make this an informative paper for the community with experiments that are comprehensible and more comprehensive.
> >
> > Here are point-by-point replies:
> >
> > > 1. The proposed setting depends on the order of domains [...] This small sampling of orders is unlikely to give a non-noisy estimate of performance
> >
> > The tables in the appendix confirm that the improvements is mostly on Digits, and not on the larger and more diverse benchmarks of PACS nor DomainNet. Furthermore, prior work on test-time adaptation has reported on more domain generalization benchmarks (see T3A, for instance) and the results would be more certain if they included these same evaluations (such as Terra Incognita or OfficeHome). Many of the tables in the appendix (12-15) still report only certain orders, which can make a reader question the overall result, which is not included by for example reporting the mean and standard error across all or a larger sampling of orders.
> >
> > > 2. The framing is neither accurate nor cordial and collegial. This paper does not begin the study of continual shift, counter to the claims of the abstract and introduction.
> >
> > This has improved, with the framing of the "unfamiliar period", but it is not appropriate to relegate related work to the appendix. The Introduction and Related Work of the main paper must clearly situate this submission. Although it can be difficult to contribute at the interface of different topics, like domain adaptation + domain generalization + continual learning, the must nevertheless be connected. For example. EATA makes clear how it connects continual learning to test-time adaptation and credits those papers.
> >
> > > 3. The evaluation protocol differs from the compared work [...]
> >
> > Thank you for including results in the established settings of the methods compared against. Why was [4] chosen as the experimental setting, when [4] does not seem to be cited in the submission? The Tables 13, 14, 15 still mostly shows improvement on Digits, and not the others, though there is improvement in TDG as claimed. Although these tables are now included, this is quite a lot to navigate for a reader, and why this review suggests that the results need to be standardized and reorganized, so that they align with prior work for rigorous comparison and so that they can be accessibly understood.

---

> > > ### Author Response · Authors · 2022-12-07
> > > **Further Response to Reviewer jYsN [1/4]**
> > >
> > > Thank you for the feedback. We are glad to see that part of your questions have been addressed, but will have to respectfully disagree on some of the new comments. We would like to provide our answers below to address them. We hope that these can help further clarify our approach and its results, and possibly help you better evaluate our work and give the final recommendation.
> > >
> > > > C1: The tables in the appendix confirm that the improvements is mostly on Digits, and not on the larger and more diverse benchmarks of PACS nor DomainNet.
> > >
> > > PACS and DomainNet are larger and more diverse than Digits, and thus achieving the same level of performance improvement could be more challenging. For example in the literature, L2D (Wang et al. ICCV 2021) achieved accuracy improvement on Digits by 15.5%, but only 4.2% on PACS, over the second-best baseline. Similarly, PCL (Yao et al. CVPR 2022), a state-of-the-art multi-domain generalization study, achieved only 0.6% performance improvement on PACS and 1.2% on DomainNet, over the second-best method. More such examples can be found in recent domain generalization works as well (DSU: Li et al. ICLR 2022, SDG: Sener et al. NeurIPS 2022). Taking these into account, while the TDG improvement of Twofer on Digits is indeed the largest (**5.6%** over the second-best method, Table 9, on average of 10 domain orders), we don’t think that the TDG improvement on PACS (**4.2%** over the second best, Table 10, on average of 10 domain orders) and DomainNet (**2.6%**, Table 11) are small at all. The improvements on TDA and FA for PACS and DomainNet are small, but TDG is our main focus.
> > >
> > > > C2: More evaluations on Terra Incognita and Office Home.
> > >
> > > We have tested Twofer and other baselines on the newly-mentioned Terra Incognita and Office Home with 4 randomly determined domain orders (more on domain orders below), as shown in the table below. According to these experiment results, Twofer is still able to generally outperform other baselines, which further demonstrates its effectiveness. We will be glad to include these results in future revision.
> > >
> > > **Office home**
> > >
> > > | Metric |   Order   | SHOT+  | SHOT++ | Tent   | AdaCon | EATA   | L2D | PDEN | Twofer |
> > > |----------|------|------|--------|--------|--------|--------|-----|------|--------|
> > > |         | ACPR | 60.0   | 57.6   | 60.3   | 60.3    | 60.3   | 56.8    | 57.2     | **63.8**   |
> > > |         | CAPR | 61.6   | 60.2   | 58.7   | 58.1    | 58.4   | 57.2    | 57.8     | **63.5**   |
> > > | TDG | PACR | 55.6   | 54.2   | 55.9   | 56.3    | 56.1   | 55.0    | 56.0     | **59.1**   |
> > > |         | RACP | 61.7   | 59.4   | 64.6   | 63.5    | 63.8   | 59.5    | 61.0     | **65.8**   |
> > > |         | Avg.    | 59.7   | 57.9   | 59.9   | 59.6    | 59.7   | 57.1    | 58.0     | **63.1**   |
> > > ||
> > > |         | ACPR | 69.5   | 68.9   | 64.7   | 67.5    | 64.2   | 65.3    | 66.0     | **69.8**   |
> > > |         | CAPR | 75.2   | 76.1   | 65.1   | 70.1    | 65.6   | 66.4    | 67.1     | **79.7**   |
> > > | TDA | PACR | 72.7   | **73.1**   | 66.0   | 68.2    | 65.6   | 66.8    | 67.9     | 72.4   |
> > > |         | RACP | 72.1   | **72.8**   | 70.9   | 70.0    | 70.5   | 70.3    | 70.2     | 71.2   |
> > > |         | Avg.    | 72.4   | 72.7   | 66.7   | 69.0    | 66.5   | 67.2    | 67.8     | **73.3**   |
> > > ||
> > > |         | ACPR | 67.1   | 62.0   | 62.7   | 62.0    | 61.4   | 60.7    | 61.5     | **68.1**   |
> > > |         | CAPR | 70.9   | 68.5   | 65.6   | 66.0    | 65.9   | 64.0    | 65.3     | **72.3**   |
> > > | FA    | PACR | 66.6   | 65.6   | 63.6   | 64.5    | 63.3   | 62.5    | 63.1     | **66.8**   |
> > > |         | RACP | 67.3   | 66.2   | 68.6   | 66.8    | 67.4   | 67.0    | 66.2     | **67.8**   |
> > > |         | Avg.    | 68.0   | 65.6   | 65.1   | 64.8    | 64.5   | 63.6    | 64.0     | **68.8**   |

---

> > > > ### Author Response · Authors · 2022-12-07
> > > > **Further Response to Reviewer jYsN [2/4]**
> > > >
> > > > **Terra Incognita**
> > > >
> > > > | Metric |  Order   | SHOT+  | SHOT++ | Tent   | AdaCon | EATA   | L2D | PDEN | Twofer |
> > > > |-----------------|---------------|--------|--------|--------|--------|--------|-----|------|--------|
> > > > |                 | 38,43,46,100  | 27.0   | 26.0   | 20.8   | 27.5   | 28.0   | 25.5    | 26.0    | **31.6**   |
> > > > |                 | 43,100,38,46  | 30.2   | 30.2   | 27.5   | 32.7   | 33.8   | 30.0    | 31.2    | **35.0**   |
> > > > | TDG        | 46,38,100,43  | 34.2   | 34.4   | 39.2   | 36.2   | 34.6   | 33.1    | 32.8    | **40.1**   |
> > > > |                 | 100,46,43,38  | 26.2   | 25.8   | 23.7   | 27.0   | 28.4   | 26.5    | 27.3    | **29.3**   |
> > > > |                 | Avg.                | 29.4   | 29.1   | 27.8   | 30.9   | 31.2   | 28.8    | 29.3    | **34.0**   |
> > > > ||
> > > > |                 | 38,43,46,100  | 46.2   | 47.2   | 39.7   | 44.7   | 44.3   | 45.3    | 45.6    | **48.7**   |
> > > > |                 | 43,100,38,46  | 38.5   | 38.2   | 44.8   | 42.9   | 45.5   | 39.0    | 40.7    | **48.9**   |
> > > > | TDA         | 46,38,100,43  | 41.8   | 43.5   | **54.4**   | 49.8   | 48.5   | 44.4    | 45.0    | 48.7   |
> > > > |                 | 100,46,43,38  | 42.6   | 44.4   | 43.8   | 44.0   | **45.8**   | 43.5    | 44.6    | 45.5   |
> > > > |                 | Avg.                | 42.3   | 43.3   | 45.7   | 45.4   | 46.0   | 43.1    | 44.0    | **48.0**   |
> > > > ||
> > > > |                 | 38,43,46,100  | 30.2   | **52.3**   | 46.3   | 45.0   | 43.8   | 40.8    | 42.0    |  47.5  |
> > > > |                 | 43,100,38,46  | 30.5   | **51.2**   | 46.7   | 42.0   | 40.3   | 37.5    | 38.6    |  46.1  |
> > > > | FA            | 46,38,100,43  | 38.9   | 53.4   | 54.8   | 45.1   | 44.2   | 42.9    | 42.2    | **55.1**   |
> > > > |                 | 100,46,43,38  | 23.6   | 50.1   | 49.3   | 43.7   | 44.2   | 37.8    | 37.9    | **58.7**   |
> > > > |                 | Avg.                | 30.8   | 51.8   | 49.3   | 44.0   | 43.1   | 40.0    | 40.2    | **51.9**   |
> > > >
> > > > > C3: Many of the tables in the appendix (12-15) still report only certain orders, which can make a reader question the overall result, which is not included by for example reporting the mean and standard error across all or a larger sampling of orders.
> > > >
> > > > We will address this issue of ordering from a few aspects:
> > > >
> > > > * First, as stated in our rebuttal summary, we tested 10 different domain orders for each dataset under the data-splitting scheme used in the original submission (the reasoning for that was also explained in the rebuttal; more on the data-splitting scheme below), and reported the results in **Tables 9-11**. Tables 12 was only showing the measurements on FT and BT (not the focus of this paper), and Tables 13-15 were for the standard data-splitting scheme. The reason why we did not also run 10 orders for the standard data-splitting scheme was that we did not have enough resources and time – as mentioned in the rebuttal, the experiments we ran were the most we could do within the rebuttal time frame (more on resources below).
> > > >
> > > > * During rebuttal, we used **all the resources we can obtain/afford** from our own GPU servers and Amazon EC2, including around 1400 GPU hours on NVIDIA RTX A5000, 200 GPU hours on NVIDIA Tesla V100, and 800 GPU hours on NVIDIA TITAN RTX. It was simply infeasible for us to run all the orders (which would be 120 for 5 domains of PACS and 720 for 6 domains of DomainNet) or anything that is significantly larger than 10.
> > > >
> > > > * Nevertheless, following your feedback, we ran **additional orders under the standard data-splitting scheme** on Digits (7 additional orders) and DomainNet (8 additional), so that they also have 10 orders in total under the standard scheme, the same as under our data-splitting scheme. In addition, we ran **all possible orders for PACS** (24 in total for the 4 domains), also under the standard data-splitting scheme. The mean and standard deviation are reported in the table below (in each cell, the left number is the mean accuracy, and the right number in the brackets is the standard deviation). The detailed results are provided in this anonymous link: https://docs.google.com/spreadsheets/d/1dT_byvIPTh1BgU2Db7z2E9J1bC9QrR0g/edit?usp=sharing&ouid=107220897664915790043&rtpof=true&sd=true. We can see that under the standard data-splitting scheme, with more orders, Twofer still shows **significant improvement in TDG** over the second-best baselines (6.0% for Digits, 5.8% for PACS, and 3.2% for DomainNet). In particular, the results for **PACS are obtained under all domain orders**. Again, these additional orders are all that we can run with our resources (thousands of GPU hours, close to what we used during the rebuttal).

---

> > > > > ### Author Response · Authors · 2022-12-07
> > > > > **Further Response to Reviewer jYsN [3/4]**
> > > > >
> > > > > * Finally, to our best knowledge, prior related works (in continual DA, test DA or source-free DA, even task-agnostic CL) carry out **significantly fewer orders than our work**. Moreover, In the 10 orders we chose in the rebuttal, we tried our best to balance the possibility of taking each domain as the starting one to increase the representativeness (Table 5 in the Appendix). During all of our experiments, we did not selectively report experimental results in any case (neither under different domain orders nor with different experiment settings). We have **provided our source code** in the supplementary materials, and promised to publicly release it after acceptance. Any reader can carry out experiments in the orders they like, and we are confident that Twofer can perform well and exceed other baseline methods.
> > > > >
> > > > > | Dataset |   Metric   | SHOT+  | SHOT++ | Tent   | AdaCon | EATA   | L2D | PDEN | Twofer |
> > > > > |------------|------|------|--------|--------|--------|--------|-----|------|--------|
> > > > > |           | TDG  | 60.0 (17.0)   | 60.7 (16.8)   | 64.3 (14.2)  | 63.3 (13.8)   | 64.0 (13.6)  | 70.9 (7.2)   | 69.7 (7.4)    | **76.9** (4.1)  |
> > > > > | Digits | TDA  | 76.0 (17.3)  | 77.6 (19.1)  | 69.0 (11.7)  | 71.6 (9.7)   | 72.0 (10.3)  | 84.3 (5.6)   | 82.3 (6.1)    | **88.7** (1.9)  |
> > > > > |           | FA     | 55.9 (18.1)  | 61.1 (17.4)  | 66.4 (17.2)  | 72.2 (11.9)   | 72.8 (11.7)  | 76.5 (4.0)   | 74.0 (4.2)    | **85.0** (2.3)  |
> > > > > ||
> > > > > |            | TDG | 55.6 (11.7)   | 56.3 (10.0)   | 65.1 (11.3)   | 64.6 (11.2)  | 64.2 (13.1)    | 63.0 (11.0)   | 62.4 (10.7)    | **70.9** (8.4)   |
> > > > > | PACS | TDA  | 84.2 (8.1)   | 86.2 (7.1)   | 77.8 (9.3)     | 79.7 (7.0)  | 80.5 (7.3)   | 82.8 (6.0)   | 82.1 (6.2)    | **86.4** (4.2)   |
> > > > > |            | FA    | 75.6 (7.8)   | 83.4 (5.5)   | 80.8 (6.8)   | 81.6 (6.6)  | 82.6 (7.0)   | 77.7 (4.5)   | 77.3 (4.5)   | **85.7** (4.8)   |
> > > > > ||
> > > > > |   | TDG | 47.3 (11.6)   | 48.2 (11.4)  | 47.3 (11.5)  | 50.9 (10.2)  | 52.1 (10.4)  | 50.7 (9.6)    | 49.3 (9.6)   | **55.3** (8.3) |
> > > > > | DomainNet | TDA | 62.0 (8.5)  | 64.4 (9.0)  | 53.5 (14.0)  | 62.2 (8.1)  | 62.5 (7.7)  | 56.2 (6.5)   | 55.6 (6.9)   | **65.8** (6.3)  |
> > > > > |   | FA    | 57.8 (8.8)  | 62.8 (5.9)  | 56.0 (15.4)  | 61.8 (9.5)  | 61.9 (8.5)  | 52.2 (9.9)   | 50.2 (10.0)   | **64.1** (6.8)  |
> > > > >
> > > > > > C4: Comments about problem framing, related works, and revision change.
> > > > >
> > > > > Thank you for acknowledging the improvement in problem framing. In terms of the related works, we have already included most of them in the Introduction, despite not having a dedicated section due to the space limitation. In future revision, we can try to condense other sections and move more comparisons with related works from Appendix to the main paper (and think of whether to have a dedicated section/subsection for related works). Nevertheless, we think that the current Introduction has highlighted the key connections and differences between Twofer and related works. More specifically, we have explained the issues of applying existing studies to tackle continual domain shifts. In particular, we have emphasized and highlighted that existing studies have poor performance before and during early adaptation on a new domain (in paragraphs 1 and 2), while the proposed Twofer can effectively improve the model generalization on a new domain and the performance of Twofer is much higher than that of state-of-the-art test-time DA during early adaptation. We have also highlighted that the generalization performance before any adaptation is very important, especially considering the cases where the domain shift is sudden and the duration of new domains is short. In this case, the performance improvement brought by test-time DA is limited since the seeing data is insufficient during early adaptation. Even after the same sufficient adaptation, Twofer can achieve better adaptation accuracy than test-time DA, and achieve comparable state-of-the-art performance as standard DA methods.

---

> > > > > > ### Author Response · Authors · 2022-12-07
> > > > > > **Further Response to Reviewer jYsN [4/4]**
> > > > > >
> > > > > > > C5: Why was [4] chosen as the experimental setting, when [4] does not seem to be cited in the submission? The Tables 13, 14, 15 still mostly shows improvement on Digits, and not the others, though there is improvement in TDG as claimed. Although these tables are now included, this is quite a lot to navigate for a reader.
> > > > > >
> > > > > > We actually directly cited the GitHub repository corresponding to [4] in the original submission since it includes the implementation details – in the Appendix, we wrote “Unlike regular domain adaptation works that use all training data for testing, we follow Wang & Lu to split 80% data as the training set and the rest 20% as the testing set for all domains of these three datasets”.  We cited [4] in the rebuttal. In future revision, we can include both and move them to the main text of the paper. As for the tables, we think that the ones included in the main paper have conveyed our main message, i.e., Twofer outperforms prior works in TDG, while providing similar TDA and FA performance. The many tables in the Appendix provide more comprehensive and more detailed results, and they further validate the same message.
> > > > > >
> > > > > > > C6: Pseudo-labeling explanation.
> > > > > >
> > > > > > Thanks for your acknowledgment. We will put the explanation in the main text and try to carry out experiments about the filtering mechanism of SHOT++ and Twofer in future revision.

---

> ### Author Response · Authors · 2022-11-18
> **Response to Reviewer jYsN [2/5]**
>
> >4. There is missing related work on test-time adaptation with more efficient updating and slower forgetting: Efficient Test-Time Model Adaptation Without Forgetting (EATA) at ICML'22. As this prior work addresses key claims of TwoFer, it needs to be compared with in the experimental evaluation. It should likewise be discussed should its contributions and technical details intersect with the claims in the proposed method.
>
> Thanks for mentioning this work. We have implemented EATA and compared it with our approach in the revision. The results are included in Sections C.1 and E of the Appendix. We can observe that our approach outperforms EATA in TDG, and achieves comparable (and in many cases better) performance in TDA and FA.
>
> >5. There is missing related work on source-free adaptation: AdaContrast (Chen et al. CVPR'22) and SHOT++ (Liang et al. PAMI'21). This requires discussion and comparison, as results are reported for DomainNet and other standard benchmarks.
>
> We also implemented AdaContrast and SHOT++ in the revision, and compared them with our approach. The results are included in Sections C.1 and E of the Appendix. We can similarly observe that our approach outperforms both AdaContrast and SHOT++ in TDG, and achieves comparable (and in many cases better) performance in TDA and FA.
>
> >6. The proposed method heavily relies on data augmentation, which makes it more domain specific than other methods, which perform none or little, like Tent or EATA for example.
>
> As mentioned in our paper, our approach Twofer starts with a pure single source domain generalization (SSDG) problem, which to our best knowledge, often relies on data augmentation to solve. In this work, we consider the domain generalization performance (i.e., TDG) as our main focus, and thus propose a novel random mixup data augmentation mechanism. We do not think that our approach is too domain-specific due to data augmentation because even without the data augmentation, the rest modules in our approach can still achieve better performance than baseline methods (please see the ablation studies of RandMix in Section 3.2 of the main text and in Section C.2 of the Appendix).
>
> **Questions about Clarity:**
>
> >1. The introduction's discussion of the "unfamiliar period" does reframe the claims in the abstract for the better, by focusing the paper on accuracy pre-adaptation and post-adaptation. However, this framing should come first, and be reflected in the abstract and first part of the introduction, before claiming that the proposed TwoFer "significantly outperforms" everything and is "envisioned as a significant milestone". These are not precise nor productive claims as written. Furthermore, in talking about the unfamiliar period and the metrics TDG, TDA, and FA, this work could align with the literature on continual learning and discuss "forward transfer" and "backward transfer" instead of introducing custom terminology.
>
> Thanks very much for this comment. We completely agree that we should have mentioned the ‘unfamiliar period’ earlier and clarify our claims. In the revision, we have reorganized and rewritten part of the Abstract and the Introduction to clarify the problem we are addressing and the corresponding contributions we made. In particular, we emphasize that our main focus is to improve the model performance on new target domains before and during their training, i.e., in the Unfamiliar Period; and the experimental results demonstrate our approach’s effectiveness in improving this objective, namely the target domain generalization (TDG) performance. We have also used target domain adaptation (TDA) and forgetting alleviation (FA) to further evaluate model performance in CDSL, and demonstrated that while not being the main focus, our approach can achieve comparable performance in these objectives (TDA and FA) as previous methods.
>
> In terms of terminology, to our best knowledge, “forward transfer” (FT) and “backward transfer” (BT) are not commonly used metrics for CDSL works. Our definitions of TDA and FA are perhaps more suitable for evaluating the model performance in CDSL since they capture the “absolute” performance rather than the relative performance change measured by FT and BT. Nevertheless, in our revision, we also calculate the FT and BT metrics as a reference and report them in the Section C.3 of the Appendix.  Finally, we have also added a discussion of related works in Section A of the Appendix to further clarify the novelty and contributions of our work.
>
> >2. The main results (Tables 1, 2, 3) obscure the compared methods with custom abbreviations (TEN instead of Tent, CoT instead of CoTTA, etc.). While these can be decoded and checked against the text, it would be faster to read if the original names and abbreviations were kept.
>
> Thanks for the suggestion. We have reorganized Tables 1-3 and changed the abbreviations back to the default.

---

> > ### Comment · Reviewer_jYsN · 2022-11-30
> > **Thank you for the point-by-point response (2)**
> >
> > > 4. There is missing related work [EATA]
> > > 5. There is missing related work on source-free adaptation [AdaCon, SHOT++]
> >
> > Thank you for including EATA, AdaCon, and SHOT++. The results for EATA, AdaCon, and SHOT++ in many cases are comparable to TwoFer, but TwoFer does improve in TDG (especially on Digits), and including these more recent and stronger baselines provides a more thorough evaluation.
> >
> > > 6. The proposed method heavily relies on data augmentation
> >
> > I grant that single domain generalization rely on augmentation, and highlighting that the proposed augmentation helps multiple methods (Figure 4a) shows that it is not too specific. This point is resolved, and not a weakness.
> >
> > > Questions about Clarity: 1. [framing]
> >
> > Thank you for the revision to focus on the unfamiliar period. This could still be improved, by highlighting where existing work does or does not measure early adaptation. For instance, online test-time adaptation methods do include early adaptation in their evaluation, since every input is scored as it is encountered, but it does not separate out metrics into TDG, TDA, and FA as done here.
> >
> > > 2. The main results (Tables 1, 2, 3) obscure the compared methods with custom abbreviations
> >
> > Thank you for changing back the default abbreviations for clarity.

---

> ### Author Response · Authors · 2022-11-18
> **Response to Reviewer jYsN [3/5]**
>
> **Questions about Quality:**
>
> >The main results in Tables 1 & 2 appear to show large gains in average accuracy across domains for the three metrics of accuracy before/during/after a given domain. However, as already noted this evaluation is not entirely standard, and the standard evaluation is not included in the results to first establish performance on a known setting.
>
> Thanks for pointing this out. In the original submission, the only difference in settings between our work and the previous work [5, 6] is that we randomly split 80% and 20% dataset for training and testing, while known settings regard the entire dataset as both training and testing sets. As explained earlier in responding to the comment on evaluation protocol, we chose our setting based on the observation from [4]. In the revision, we have added experiments based on the known setting in previous work [5, 6], and presented the results in Section E of the Appendix (Tables 13, 14, 15). From the results, we can observe that our approach still works effectively and outperforms baselines in TDG, while achieving comparable performance in TDA and FA.
>
> [4] Wang J, et al. Generalizing to unseen domains: A survey on domain generalization. TKDE 2022.
>
> [5] Shiqi Yang, et al. Generalized source-free domain adaptation. ICCV 2021.
>
> [6] Dian Chen, et al. Contrastive test-time adaptation. CVPR 2022.
>
> **Questions about Novelty:**
>
> >1. Contrary to its claim, Twofer is not the first work to address continual adaptation or "continual domain shift learning". Efficient Test-Time Model Adaptation without Forgetting at ICML'22 and Continual Test-time Domain Adaptation CVPR'22 both address adaptation to a sequence of domains, did so well before the submission deadline for ICLR'23, and report results on larger-scale and more difficult datasets. The earliest instance of learning on non-stationary domains is most likely Continuous Manifold Based Adaptation for Evolving Visual Domains, although it works in the unsupervised domain adaptation setting, but that does not make it unrelated.
>
> Thanks for the comment. We did not mean to claim that our work is the first to address continual adaptation and we apologize for the confusion. In the revision, we have made changes throughout the paper, particularly in the Abstract and the Introduction, to clarify the focus and novelty of our approach. In particular, we emphasize that our main focus is to improve the model performance on new target domains before and during their training, i.e., in the unfamiliar period. We also added a discussion of related works in Section A of the Appendix to further clarify the relation and differences between our work and previous ones.
>
> >2. The proposed pseudo-labeling approach is in essence clustering while filtering the predictions by certainty, but such filtering or thresholding is common for pseudo-labeling. See the PAMI edition of SHOT (Liang et al. 2021) called SHOT++ or EATA (cited above in this review) for examples. In particular SHOT++ has an adaptive threshold based on the distribution of confidences, which is related to filtering a fixed percentage as done by T2PL.
>
> Thanks for pointing out these related works. While we use a filtering strategy for cluster centroids construction in the pseudo labeling process, SHOT++ filters samples for the semi-supervised learning process and it still takes all samples into account when constructing cluster centroids in the pseudo labeling process. Besides, SHOT++ still relies on one-to-one distance between unlabeled samples and cluster centroids to allocate pseudo labels. However, in cross-domain learning, samples from different domains are highly overlapped, and allocating pseudo labels by comparing samples with centroids of different classes only once is risky. In our kNN-based pseudo labeling, kNN can decentralize the risk of misclassification in assigning pseudo labels caused by a single comparison between class centroids and data samples. Besides, our T2PL works as a voting system that can encourage the features of certain samples to have the same labels as their neighbors and can be interpreted as a kind of local structure clustering as mentioned in G-SFDA [5]. As a result, T2PL is more suitable for clustering samples that are highly overlapped than center-based approaches. Moreover, the filtering strategy of T2PL is much simpler than that of SHOT++, showing its difference from SHOT++ and other similar pseudo-labeling approaches in both sample filtering and clustering.

---

> > ### Comment · Reviewer_jYsN · 2022-11-30
> > **Thank you for the point-by-point response (3)**
> >
> > > Questions about Quality: [...] evaluation is not entirely standard
> >
> > Thank you for including a more standardized evaluation. However, it is not so informative to have the standard, comparable evaluation all the way in the back of the appendix and not in the main paper. Reorganization is needed to make the experimental settings clear and communicate which results are comparable and how.
> >
> > > We chose our setting based on the observation from [4]
> >
> > But [4] is not referenced in the submission? (Unless I missed it, in which case disregard this comment, of course).
> >
> > > Questions about Novelty: 1. Contrary to its claim, Twofer is not the first work to address continual adaptation
> >
> > Thank you for revising this claim. However, adding more discussion of related work to an appendix does not mean that the setup and contribution of twofer will be clear from reading the main paper. I advise editing the Introduction, and considering whether a separate Related Work section is needed, to better highlight the claimed missing piece in existing research (accuracy before adaptation/during early adaptation), and explain how this is not addressed by the nearby topics (DG, DA, CL).
> >
> > > 2. The proposed pseudo-labeling approach is in essence clustering while filtering
> >
> > Thank you for the further discussion of the method details. To make this point rigorously, the paper should include this explanation of how T2PL differs from prototypical methods, and ideally the experiments would even measure the amount of overlap to identify where/when T2PL improves on the filtering of SHOT++.

---

> ### Author Response · Authors · 2022-11-18
> **Response to Reviewer jYsN [4/5]**
>
> >3. The use of parameters as prototypes is not novel to this work, and has been done not only by the cited Saito et al. 2019, but goes back to at least Imprinted Weights by Qi et al. CVPR'18.
>
> While there are previous works in using parameters as prototypes, we think that our usage of classifier-based prototypes has its novelty and is pertinent to the problem we address. Specifically, we have explained our motivation for adopting prototype learning to tackle continual domain shifts (cross-domain sample variability and sample insufficiency, corresponding to the randomness of RandMix data augmentation and a small set of stored old samples). Existing prototype construction methods have two strategies: one based on aggregating forwarding representations and the other on viewing classifier parameters as prototypes. The novelty of our work is not so much on prototype construction, but on how to use and optimize such prototypes. In this case, we clearly describe the shortcomings of applying forwarding-based prototypes to our problem, especially considering the adaptivity gap expanding. Then we design a new contrastive comparison-based alignment method to better use and optimize classifier-based prototypes. We believe that this is well-motivated, and the proposed prototype learning algorithm indeed works effectively in solving our problem.
>
> **Rebuttal Major Questions:**
>
> >1. Please report results in the standard evaluation protocols for PACS and DomainNet. In particular, please use the standard splits, and include results with stationary/non-continual/episodic domains. By comparing in the established setting, the experiments would establish that Twofer is at least as good there, and then the experiments could should improvement in the proposed protocol.
>
> Thanks for the suggestions. As mentioned above, we have conducted new experiments for the standard data splitting setting used in previous works [5, 6], and also for the stationary domain adaptation setting. The results are included in Section E of the Appendix (Tables 13-17), and demonstrate that our approach still outperforms baselines in TDG, while achieving comparable TDA and FA.
>
> >2. Please analyze the sensitivity of TwoFer and the baselines to domain order. If the variance due to order is larger than the reported gains, then the experiments may have inadvertently chased noise.
>
> Thanks for the suggestion. We have carried out additional experiments to assess the impact of domain order on Twofer and baselines, and report the results in Section C.1 of the Appendix. Specifically, we randomly shuffle the domains for all three datasets and try to balance the possibility of taking each domain as the starting one to increase the representativeness (Table 5 in the Appendix). We conduct experiments for 10 random orders for each dataset (the most we can do within the rebuttal time frame), and report the average TDG, TDA, and FA under these 10 domain orders. The experimental results demonstrate similar observations as in the original submission, i.e., Twofer significantly outperforms state-of-the-art baseline methods in TDG, and achieves comparable performance in TDA and FA. Please refer to Tables 9, 10 and 11 in the Appendix for detailed results.
>
> >3. Please relate TwoFer to EATA and AdaContrast and report any comparisons, if any comparable comparisons are possible given the set of experiments in the submission.
>
> We have implemented EATA, AdaContrast, and SHOT++, and compared them with Twofer. The results are reported in Sections C.1 and E of the Appendix (Tables 9-11 with the same data-splitting setting as in the original submission, Tables 13-15 with the standard data-splitting setting as in [5, 6], Tables 16 and 17 for stationary cases). Across these different cases, our approach Twofer shows that it can outperform baselines in TDG, while achieving comparable performance in TDA and FA.

---

> ### Author Response · Authors · 2022-11-18
> **Response to Reviewer jYsN [5/5]**
>
> >4. Please clarify the novelty or lack thereof in the setting, compared to the missing related work raised, and re-articulate the contributions of CDSL vs. other studies of continual shift in existing work like CoTTA and EATA.
>
> Thanks for the comment. In the revision, we have made changes throughout the paper, particularly in the Abstract and the Introduction, to clarify the focus, novelty, and contributions of our approach. In particular, we emphasize that our main focus is to improve the model performance on new target domains before and during their training, i.e., in the Unfamiliar Period; and the experimental results demonstrate that our approach Twofer can indeed significantly improve this objective, namely the target domain generalization (TDG) performance, over previous state-of-the-art methods. In addition, while not being the main focus, our approach can also achieve comparable performance as those previous methods in model target domain adaptation (TDA) and forgetting alleviation (FA) capabilities. This balanced performance across TDG, TDA, and FA will hopefully help the adoption of our approach.
>
> We have implemented additional baseline methods for comparison (EATA, SHOT++, AdaContrast), explored more domain orders, and added experiments for standard data-splitting settings and stationary domain adaptation. The additional experimental results consistently demonstrate the effectiveness of our approach in improving TDG while balancing TDA and FA. We also added a discussion of related works to further clarify the novelty of our work.
>
> >5. Was DomainNet filtered according to the evaluation protocol in prior work like Saito et al. 2019 and Chen et al. 2022 (AdaContrast)?
>
> No, it was filtered differently. The evaluation protocol in AdaContrast obtains only 4 domains for DomainNet, which we think is similar to PACS in terms of domain numbers. Instead, we follow the evaluation protocol in references [7] to filter DomainNet.
>
> [7] Xie, et al. General incremental learning with domain-aware categorical representations. CVPR 2022

---

> ### Author Response · Authors · 2022-11-21
> **A Gentle Reminder of Further Feedback**
>
> Thank you again for the first round of review feedback. We hope our response is able to address your comments related to the sensitivity of domain orders, problem framing, further comparison with more baselines, and performance on more experiment settings. We take this as a great opportunity to improve our work and shall be grateful for any additional feedback you could give us.

---

> ### Author Response · Authors · 2022-11-29
> **With the hope that our response addresses your concerns**
>
> Dear Reviewer jYsN,
>
> The conclusion of discussion period is closing, and we eagerly await your response. We greatly appreciate your time and effort in reviewing this paper and helping us improve it.
>
> We have provided detailed responses to every one of your concerns. Please help us to review our responses once again and kindly let us know whether they fully or partially address your concerns and if our explanations are in the right direction. We shall be grateful for any additional feedback you could give us.
>
> Kind Regards,
>
> Authors of Paper477

---

### Official Review · Reviewer_Nus3 · 2022-10-30

**Confidence:** 3
**Correctness:** 4
**Technical Novelty And Significance:** 3
**Empirical Novelty And Significance:** 3
**Recommendation:** 8

**Clarity, Quality, Novelty And Reproducibility:**

The paper is well-organized and clearly written, and it focuses on practical issues in deep learning model deployment. This work should be reproducible. The implementation details and code are provided.

**Strength And Weaknesses:**

Strength:

1. The objective of target domain generalization is appealing and significant, and I particularly like the definition of ‘unfamiliar period’ in the paper. Although a number of domain adaptation studies have been proposed, including continual and source-free, most of them neglect the generalization before adaptation. This paper bridges this gap by allowing the deployed models to evolve themselves and remain stable performance all the time.

2. The proposed Twofer is composed of three major modules, which work together for achieving better domain alignment. These three modules are well-motivated.

3. The author(s) compare twofer with a number of SOTA baselines. The experiments are extensive, and the results can demonstrate the effectiveness of proposed methods.

Weakness:

1. If I understand correctly, twofer needs to store a small number of samples at each stage for later usage, is it possible to relax this requirement? More should be discussed.

2. I notice that there is no section of related work, which is important to this work. Therefore, I believe the author(s) needs a section or a table to present the difference among different DA and DG topics.

Questions:

1.	The author(s) mentions that blindly pushing augmentation data away from the original possibly hurts the generalization performance. Could the author(s) give more explanation here?

2.	Besides, is it possible to extract useful information from seen domains for guiding the data augmentation in the future? Because the used augmentation module is simple in terms of network structure, the training cost is acceptable if learnable augmentation can bring noticeable performance gain.


**Summary Of The Paper:**

This work focuses on a practical challenge that deep learning models cannot remain stable performance when being deployed in real-world environments. For this problem, the author(s) defines three objectives, i.e., target domain generalization, target domain adaptation and forgetting alleviation. To achieve these objectives, a framework called Twofer is proposed to better align domains over time. According to extensive evaluation experiments and comprehensive baseline comparison, the effectiveness of Twofer can be demonstrated.

**Summary Of The Review:**

This paper is clear and technically solid. The experiments are thorough with impressive results, and the analysis is extensive.

---

> ### Author Response · Authors · 2022-11-18
> **Response to Reviewer Nus3**
>
> >1. If I understand correctly, twofer needs to store a small number of samples at each stage for later usage, is it possible to relax this requirement? More should be discussed.
>
> Thanks for the question. Yes, Twofer stores a small number of old samples for the forgetting compensation. These stored samples are put into a data memory that has a fixed size, which means that the number of stored samples will not linearly increase when seeing more and more domains. As for relaxing this requirement, we think it is possible. For example, we may obtain a prototype of each class from the learned classifier at each stage and design a special data generation module to generate samples that are close to such prototypes. We believe that this design can be incorporated into our current data augmentation module, and we plan to explore this in future work.
>
> >2. I notice that there is no section of related work, which is important to this work. Therefore, I believe the author(s) needs a section or a table to present the difference among different DA and DG topics.
>
> Thanks for the suggestion. We have added a discussion of related works, including a table for summarizing the differences among various topics, in Section A of the Appendix.
>
> >3. The author(s) mentions that blindly pushing augmentation data away from the original possibly hurts the generalization performance. Could the author(s) give more explanation here?
>
> In state-of-the-art single-source domain generalization works, e.g., L2D (Wang et al. ICCV 2021) and PDEN (Li et al. CVPR 2022), the data augmentation modules are usually learnable. Furthermore, the learning of data augmentation is conducted in an adversarial training way, i.e., the feature encoder tries to extract similar representations from both the original and the augmented data while the data augmentation module aims to generate data that is as different from the original as possible. As the training goes on, the semantic distance between the augmented and the original data becomes further and further, which also means that the shared semantics between them become less and less. Although the feature encoder’s ability to extract shared semantics is getting stronger over training time, the survived shared semantics may be too few to support good generalization performance in the target domains. From our experiments, we observed that stopping the training of the data augmentation module in L2D and PDEN early can help improve performance. As a result, we think that blindly pushing augmentation data away from the original may hurt the generalization performance in target domains.
>
> >4. Besides, is it possible to extract useful information from seen domains for guiding the data augmentation in the future? Because the used augmentation module is simple in terms of network structure, the training cost is acceptable if learnable augmentation can bring noticeable performance gain.
>
> Thanks very much for the question. Indeed for future work, our plan is to first improve the data augmentation module, and we have an idea that considers useful information from seen domains. Specifically, after training on certain old domains, the data augmentation for the next domain has possibilities – one is that the next domain is similar to one of the seen domains, while the other is that the next domain is quite different from all seen domains. Thus, we can control the data augmentation module to generate data that is in the same directions as the seen domains but further. While for the rest unexplored directions, we also need to augment sufficient data. And it may be better to augment data that is not further than the furthest distance between the current and seen domains, for preventing the lack of sufficient shared semantics.

---

> > ### Comment · Reviewer_Nus3 · 2022-11-20
> > **Response to authors**
> >
> > Thanks for addressing my comments and questions. I will keep my original score.

---

> > > ### Author Response · Authors · 2022-11-21
> > > **Thanks to Reviewer Nus3**
> > >
> > > Thank you very much for checking our response and thanks again for the initial comments. They are very helpful for our improvement of the paper.

---

### Author Response · Authors · 2022-11-18
**Thanks for All Reviewers**

We would like to thank all the reviewers for their insightful comments and constructive suggestions. Below we provide our detailed response to each reviewer. We have uploaded a revised version of our submission, with major changes highlighted in magenta (there are also other minor changes in wording and typos). New experiments are also conducted, and the results are added to the tables in magenta color. If an entire table is new (Tables 5 and 9-17 in the Appendix), only its caption is marked in magenta for readability. Thank you and we look forward to further feedback and discussion.

---

### Author Response · Authors · 2022-11-18
**Revision Change Summary**

1. [Abstract, Introduction] We have reorganized and rewritten part of the Abstract and the Introduction to clarify the problem we are addressing and the corresponding contributions we made. In particular, we emphasize that our main focus is to improve the model performance on new target domains before and during their training, in what we call the “Unfamiliar Period”; and the experimental results demonstrate that our approach can indeed significantly improve this objective, namely the target domain generalization (TDG) performance, over previous state-of-the-art methods. In addition, while not being the main focus, our approach also achieves comparable performance as those previous methods in model target domain adaptation (TDA) and forgetting alleviation (FA) capabilities.

2. [Section 3.1, Tables 1-3] We have carried out additional experiments with new settings for two baseline methods, SHOT and BMD, to further compare them with our approach. In the original submission, we set exactly the same training iterations for those two baselines as our approach. However, we later found that SHOT and BMD perform better when loading their default settings of the training iterations. Thus in the revision, for a fair comparison, we report experimental results with the default training iterations for SHOT and BMD (Tables 1-3 in the main text). Our approach still significantly outperforms the two in TDG, and achieves comparable performance in TDA and FA.

3. [Appendix, Section A] We have added a discussion of relevant works to our approach, including methods in Fine-Tuning, Continual Learning, Unsupervised DA, Continual DA, Source-Free DA, Test-Time DA, Single Source DG, Multi-Source DG, and Unified DA & DG. Please refer to Section A of the Appendix for details.

4. [Appendix, Sections C.1 and E] We have considered three more state-of-the-art baseline methods, SHOT++ (Liang et al. TPAMI 2021), AdaCon (Chen et al. CVPR 2022), EATA (Niu et al. ICML 2022), and compared them with our approach. The results are included in Sections C.1 and E of the Appendix, and still demonstrate the advantage of our approach in improving TDG and providing comparable TDA and FA.

5. [Appendix, Sections C.1 and C.2] We have carried out additional experiments with more domain orders for assessing their impact on performance. The results are included in Section C.1 of the Appendix (Tables 9-11), and similarly, demonstrate the effectiveness of our approach. We have also conducted new ablation studies on PACS and DomainNet datasets (in addition to the ablation studies reported in the main text on Digits). The results are also included in Section C of the Appendix (Figures 5 and 6). For these additional experiments, to save time and space, we select a subset of baseline methods that perform relatively better than the rest for evaluation, including the new baseline methods mentioned above.

6. [Appendix, Section C.3] We provide the performance evaluation measured by forward and backward transfer on main experiments. Please refer to Section C.3 (Table 12) for detailed results.

7. [Appendix, Section E] We have conducted new experiments for the standard data splitting setting of domain adaptation. While such a setting is common in the DA works, we think it may not be practical enough in CDSL. Nevertheless, the experimental results demonstrate that under this setting, our approach can still outperform baselines in TDG while achieving comparable TDA and FA.  We also conducted new experiments for the stationary domain adaptation setting, and the results demonstrate the effectiveness of our approach. Please refer to Section E of the Appendix for detailed results of these two sets of experiments (Tables 13-17).

---

### Decision · Program_Chairs · 2023-01-20

**Decision:**

Accept: poster

**Justification For Why Not Higher Score:**

Although the studied problem is new and challenging, reviewers have minor concerns on the technical contributions of the proposed method.

**Justification For Why Not Lower Score:**

This paper investigates a new problem, continual domain shift, and presents a novel solution. It will inspire other researchers in this field.

**Metareview: Summary, Strengths And Weaknesses:**

This paper investigates a new and challenging research problem, i.e., continual domain shift, and presents a novel framework named Twofer to address it. Unlike existing work, Twofer is able to simultaneously achieve target domain generalization (TDG), target domain adaptation (TDA), and forgetting alleviation (FA). Extensive experimental results on multiple benchmark datasets are reported and discussed.

Overall, this paper is well written and clearly organized. The paper studies a practical and challenging research problem. The motivation of the proposed Twofer framework is clear, and the technical details of this paper are easy to follow. Comprehensive experiments in various settings demonstrate the effectiveness of the proposed method over baselines.

Meanwhile, reviewers raised some comments such as technical contribution, model justification, and experiments. The authors provided very detailed responses as well as additional results, which addressed many of the previous concerns from reviewers. During the discussion phase, reviewers recognized the major contributions of this work, especially the investigation on continual domain shift, and also suggested that the authors should incorporate the new results and discussions to the final version.

**Note From Pc:**

if the above contains the word "oral" or "spotlight" please see: "oral" presentation means -> notable-top-5% and "spotlight" means -> notable-top-25%. As stated in our emails, we are disassociating presentation type from AC recommendations

**Summary Of Ac-Reviewer Meeting:**

During the AC-reviewer meeting, reviewers recognized the major contributions of this work, especially the investigation of a new problem, i.e., continual domain shift. Meanwhile, reviewers had minor concerns on the technical contributions of the proposed method.
Considering the main contribution of this work (i.e., a new research problem) would inspire other researchers in the community, the AC and most reviewers recommended the acceptance of this paper.